# Time-series visual representations for sleep stages classification

Rebeca Padovani Ederli[ORCID][1][*], Didier A. Vega-Oliveros[2][☾], Aurea Soriano-Vargas[ORCID][3][☾], Anderson Rocha[1][‡], Zanoni Dias[ORCID][1][‡]

**1** Institute of Computing, University of Campinas (Unicamp), Campinas, SP, Brazil, **2** Department of Science and Technology, Federal University of Sao Paulo (Unifesp), São José dos Campos, SP, Brazil, **3** Departamento Académico de Ciencia de Computación y Datos, Universidad de Ingeniería y Tecnología (UTEC), Peru

‡ These authors also contributed equally to this work.
☾ These authors contributed equally to this work.
* rebeca@ic.unicamp.br

## Abstract

Polysomnography is the standard method for sleep stage classification; however, it is costly and requires controlled environments, which can disrupt natural sleep patterns. Smartwatches offer a practical, non-invasive, and cost-effective alternative for sleep monitoring. Equipped with multiple sensors, smartwatches allow continuous data collection in home environments, making them valuable for promoting health and improving sleep habits. Traditional methods for sleep stage classification using smartwatch data often rely on raw data or extracted features combined with artificial intelligence techniques. Transforming time series into visual representations enables the application of two-dimensional convolutional neural networks, which excel in classification tasks. Despite their success in other domains, these methods are underexplored for sleep stage classification. To address this, we evaluated visual representations of time series data collected from accelerometer and heart rate sensors in smartwatches. Techniques such as Gramian Angular Field, Recurrence Plots, Markov Transition Field, and spectrograms were implemented. Additionally, image patching and ensemble methods were applied to enhance classification performance. The results demonstrated that Gramian Angular Field, combined with patching and ensembles, achieved superior performance, exceeding 82% balanced accuracy for two-stage classification and 62% for three-stage classification. A comparison with traditional approaches, conducted under identical conditions, showed that the proposed method outperformed others, offering improvements of up to 8 percentage points in two-stage classification and 9 percentage points in three-stage classification. These findings show that visual representations effectively capture key sleep patterns, enhancing classification accuracy and enabling more reliable health monitoring and earlier interventions. This study highlights that visual representations not only surpass traditional methods but also emerge as a competitive and effective approach for sleep stage classification based on smartwatch data, paving the way for future research.

**Data availability statement:** The data underlying the results presented in the study are

available from PhysioNet (https: //physionet.org/content/sleep-accel/1.0.0/).

**Funding:** Part of the results presented in this work was obtained through the project "Hub of Artificial Intelligence in Health and Wellbeing - Viva Bem," funded by Samsung Eletrônica da Amazônia Ltda., within the scope of the Information Technology Law 8.248/91. There was no additional external funding received for this study. The funders had no role in study design, data collection and analysis, decision to publish, or preparation of the manuscript.

**Competing interests:** The authors have declared that no competing interests exist.

## Introduction

The sleep stages are essential for maintaining health and diagnosing sleep disorders. According to the guidelines of the American Academy of Sleep Medicine (AASM), sleep stages are classified into five distinct categories: wake, NREM N1, NREM N2, NREM N3, and REM, each with specific physiological characteristics [1]. Sleep cycles, composed of NREM and REM stages, occur every 90 to 120 minutes throughout the night and are critical for restorative sleep. Alterations in these patterns often indicate the presence of sleep disorders [2]. To simplify analysis, some studies group these stages into four (wake, light sleep, deep sleep, and REM), three (wake, NREM, and REM), or two stages (wake and sleep) [3].

Polysomnography (PSG) is the gold standard for evaluating sleep stages in clinical settings, providing detailed data from multiple physiological signals. However, PSG presents notable challenges, including high equipment costs, patient discomfort, and the need for monitoring in controlled environments, which may disrupt natural sleep [4]. Smartwatches have emerged as a practical, less invasive alternative for home sleep monitoring. These devices, equipped with sensors such as accelerometers and heart rate monitors, enable continuous data collection, promoting habit adjustments and health improvements [5].

Sensor data, often organized as time series, have been widely used in various sleep-related applications [6–8]. A common approach involves extracting features from these time series and using classical machine learning algorithms like Support Vector Machine (SVM), K-Nearest Neighbors (KNN), and Random Forest (RF). With advancements in deep learning, two prominent methodologies for processing raw data have gained attention. The first uses Recurrent Neural Networks (RNNs), such as Long Short-Term Memory (LSTM) [9] and Gated Recurrent Unit (GRU) [10], which are designed to capture temporal patterns and long-term dependencies. The second employs 1-dimensional convolutional neural networks (1D-CNNs), effective for learning local patterns in sequential data.

Studies have explored various devices and approaches to classify sleep stages. Single sensors, such as photoplethysmography (PPG), have been used to extract features and achieve promising results with classifiers like SVM [11]. Other studies combined wavelets with RF, revealing that variables like age and sleep periods influence performance [12]. Smartwatches equipped with accelerometers and PPG have demonstrated their potential as viable PSG alternatives by applying recurrent neural networks for classification [13]. Additionally, methods directly processing raw data with models such as LSTMs have shown success in analyzing activity and heart rate data [14]. Deep networks trained on multimodal PSG data further highlight the potential of end-to-end learning directly from raw inputs [15].

Despite advancements, existing approaches have limitations. Manual feature extraction often requires domain expertise, is sensitive to noise, and fails to capture complex relationships within the data [3,16]. In contrast, raw data processing struggles with high dimensionality, noise, and difficulty identifying temporal and spatial patterns, leading to compromised interpretability [3,14,15].

The rise of 2D-CNNs has introduced a promising alternative. These networks are designed to detect local patterns, such as edges, textures, and shapes, making them highly effective for computer vision tasks. Pooling layers reduce data dimensionality while preserving crucial features, and the hierarchical structure of 2D-CNNs enables the analysis of visual patterns in two-dimensional data [17].

Transforming time series into visual representations has proven to be an effective technique for sensor data analysis. This approach converts one-dimensional time series into two-dimensional images, allowing the direct application of deep learning models. Representations like Recurrence Plots (RP) [18], Gramian Angular Fields (GAF) [19], Markov Transition

Fields (MTF) [19], and spectrograms capture diverse aspects such as temporal features, state transitions, and phase space representations [20,21].

Image-based methods have achieved remarkable results in human activity recognition (HAR). For instance, integrating GAF, RP, and MTF into a convolutional model enhanced gymnastics action recognition [22]. Similarly, spectrograms of inertial and biological signals have improved the classification of activity intensity levels [23]. Gesture recognition studies have also utilized GAF and MTF, achieving high accuracy in classifying wrist movements related to food intake [24].

Although the transformation of smartwatch data into images is established in fields like HAR, its application in sleep stage classification remains underexplored. Addressing this gap, this study aims to classify sleep stages by leveraging smartwatch sensor data transformed into visual representations and applying deep learning techniques. The methodology uses the publicly available Sleep Accel dataset [25], containing accelerometer and heart rate data from Apple Watch devices, annotated with PSG-based sleep stages. Time series data were transformed into image representations, including RP, GAF, MTF, and spectrograms, to capture temporal and spatial patterns. Additionally, images were divided into patches, enabling the classification models to focus on local details. These models, combined with ensemble techniques, demonstrated improved prediction accuracy. Performance was evaluated using balanced accuracy and Cohen's kappa coefficient. Comparisons with traditional methods, such as raw data models and feature extraction approaches, highlighted the advantages of visual representations.

This study addresses two distinct classification tasks: (1) two-stage classification (sleep/wake) and (2) sleep stages classification (wake/NREM/REM). The primary goal of binary classification is to distinguish between wake and sleep periods, making it useful for applications that require basic sleep detection, such as large-scale monitoring of sleep-wake patterns or the initial assessment of sleep disorders. On the other hand, the sleep stages classification aims to identify more detailed patterns by separating sleep into NREM and REM stages, which is essential for more in-depth clinical analyses, such as detecting specific sleep disorders (e.g., sleep apnea or insomnia) and assessing sleep quality based on the sleep-wake cycle architecture.

The transformation of time series into visual representations, combined with deep learning techniques, has proven effective for both classification tasks, highlighting the versatility of the proposed methodology. This strategy advances the understanding of sleep patterns, representing a promising avenue for future research.

## Materials and methods

This section describes the publicly available database, methodology, and evaluation metrics adopted in this work.

### Dataset

The Sleep Accel dataset [25], collected with Apple Watch (series 2 and 3, *Apple Inc.*), is the most suitable publicly available dataset to date. It was collected at the University of Michigan between June 2017 and March 2019 and contains data from 31 subjects. The data includes step count, acceleration, heart rate recorded with the Apple Watch, and sleep stages labels, scored with PSG recordings according to AASM's standards. The time information (in seconds from the start of the PSG) is provided for each data point.

The Apple Watch uses a triaxial MEMS accelerometer, which measures acceleration in the $x$, $y$, and $z$ directions (in $g$), and photoplethysmography (PPG) on the dorsal wrist, which

obtains heart rate in beats per minute (bpm). The study uniquely utilized the Apple Watch for seven to 14 days, with participants spending the final night in a sleep lab for an 8-hour PSG recording while wearing the smartwatch. Notably, the participants in this study did not have any known sleep disorder diagnoses, ensuring the reliability of the data.

It is important to note that this dataset is imbalanced, considering the two-stage classification (sleep and wake) and the sleep stages classification (wake, NREM, and REM). The class proportions are 12 "Sleep" samples for each "Wake" sample (sleep/wake classification), and nine "NREM" samples for every three "REM" samples and each "Wake" sample (sleep stages classification). The class proportions and imbalance are visible in the graph in Fig 1, which shows the percentage of data for each class every hour.

In addition to the temporal distribution of labels shown in Fig 1, Table 1 provides a detailed breakdown of the dataset, presenting the absolute number of samples per class and their respective proportions. The data confirms the class imbalance, particularly in the two-stage classification, where the number of "Sleep" samples is significantly higher than "Wake" samples, and in the sleep stages classification, where "NREM" is the most frequent class.

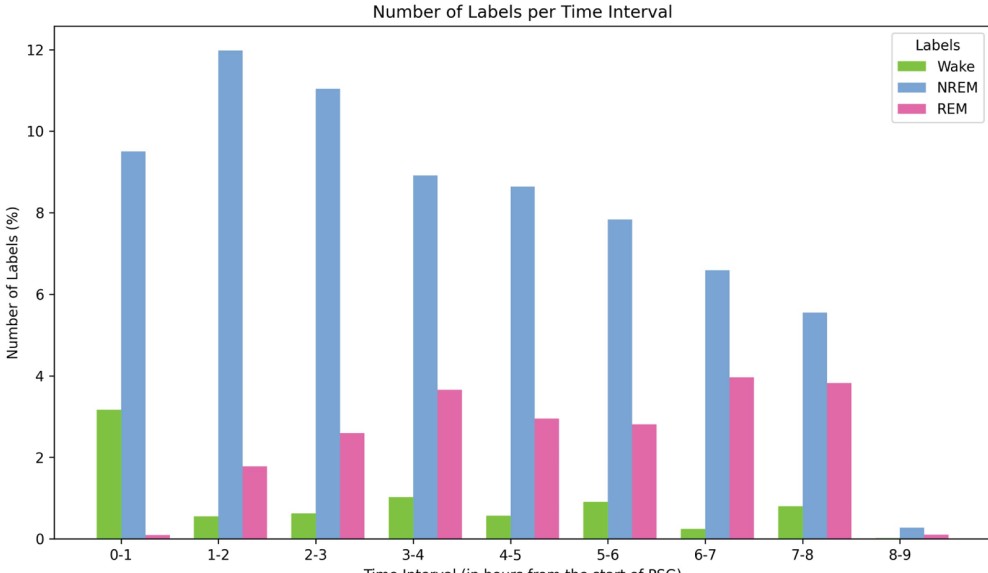

**Fig 1. Label distribution in sleep accel dataset**. Number of labels in the Sleep Accel dataset per time interval over 8 hours (considering three sleep stages). The dataset shows class imbalance, with 9 "NREM" labels for every 3 "REM" labels and 1 "Wake" label.

**Table 1**. **Distribution of samples per class for each classification task**. The percentages indicate the proportion of each class in relation to the total dataset.

| Classification | Class | Number of samples | Percentage (%) |
|---|---|---|---|
| **Two-stage** | Wake | 1935 | 7.7% |
| | Sleep | 23364 | 92.3% |
| **Sleep stages** | Wake | 1935 | 7.7% |
| | NREM | 17811 | 70.4% |
| | REM | 5553 | 21.9% |

## Overview of visual representations for sleep stages classification

A time series is a set of observations collected sequentially over time, according to a specific sampling rate. In this work, time series are obtained for the entire duration of the sleep recording, with sampling rates varying according to the sensor used. These data, which may have timestamps in different formats, are analyzed in periods of 30 seconds, called "epochs". The time series of PSG recordings are standardized into 30-second epochs, each with a corresponding label for a sleep stage. Accelerometer data from smartwatches are also time series, measuring the device's acceleration over time. The start and end times of the time series from different types of data (smartwatch and PSG) must be synchronized.

This paper investigates using visual representations of smartwatch data to classify sleep stages. The approach involves methods that incorporate both the spatial and temporal aspects of the data, integrating accelerometer (ACC) and heart rate (HR) information from smartwatches. The classification of sleep stages was simplified into classifying sleep/wake (binary classification) and wake/NREM/REM (sleep stages classification). Notably, the data are analyzed in real-time, meaning only the data available up to the current moment are considered, demonstrating the practical application of the research.

To provide a high-level overview of the proposed method, Fig 2 illustrates the general pipeline followed in this study. The process starts with raw time series data from smartwatches, which are transformed into visual representations. The final output consists of predictions for two scenarios: two-stage (wake/sleep) and sleep stages classification (wake/NREM/REM).

A more detailed methodology breakdown is presented in Fig 3, where the illustrated process is applied to both classification scenarios. The transformation of raw data into images was applied using techniques found in some related works: RP, GAF, MTF, and spectrograms. The images were generated from ACC data and HR rate data separately, and ensemble techniques were also performed to combine the classifications obtained with each data type.

Using images to perform classification tasks allows the application of a technique that divides the original image into sub-images or patches. As shown in Fig 3, the images of different representations (RP, GAF, MTF, and spectrograms), in addition to being processed in the format in which they were initially generated, are also divided into patches. Ensembles are applied to combine the predictions of the models trained with the following inputs: original ACC and HR data, patches of ACC data, patches of HR data, and patches of ACC data and patches of HR data.

The following subsections present the details of the methodology regarding data preparation, representations, patches, and ensembles.

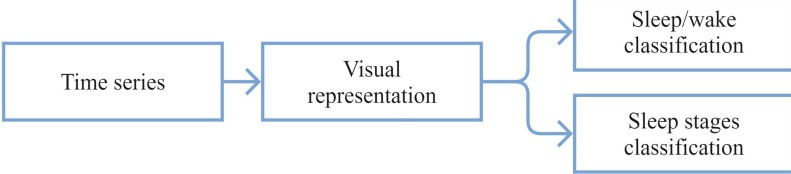

**Fig 2. General pipeline of the proposed method.** The process begins with time series data, which are transformed into visual representations. The final output consists of predictions for two scenarios: two-stage (wake/sleep) and sleep stages classification (wake/NREM/REM).

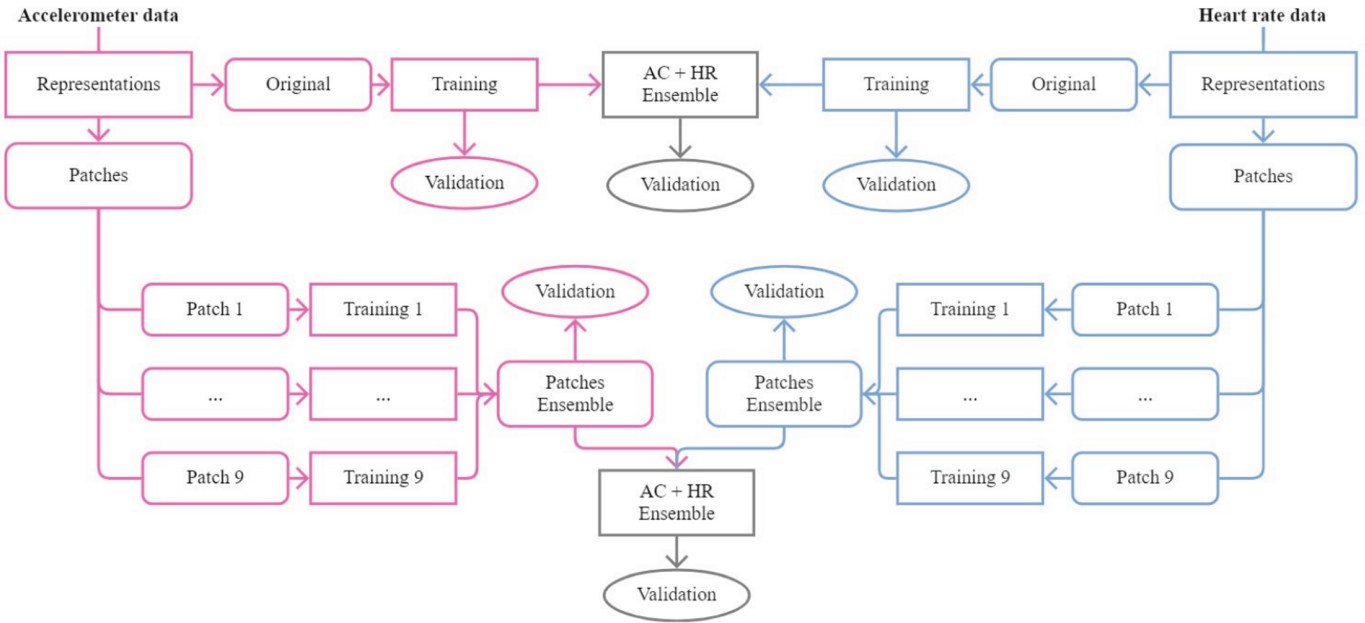

**Fig 3. Proposed methodology scheme.** The process begins with transforming raw accelerometer (ACC) data (pink) and heart rate (HR) data (blue) into visual representations. These visual representations serve as the input for training and validation. An ACC and HR data ensemble is created and validated (gray). The images are divided into patches used as inputs for their respective training sessions. Validation is carried out based on the ensemble results obtained from all patches. Finally, the ACC + HR ensemble is performed and validated again after obtaining the ensemble results from the patches (gray).

## Data preparation

A measurement refers to a single data point collected from sensors. These measurements form a time series, a sequence of data points recorded at regular intervals. Before generating the images, the raw motion and heart rate data were interpolated. Each image was created using 600 measurements [25], resulting in 600×600 pixels, except for spectrograms, with varied dimensions. For processing as input into a network, the images are resized to 224×224 pixels.

Each image corresponds to a 30-second window by the PSG recording standard used in the literature. Therefore, each image is classified as sleep or wake in the binary classification problem and as Wake, NREM, or "REM" in the three-stage classification.

The different image representations were obtained for each accelerometer axis ($x$, $y$, and $z$), and an RGB image combining $x$, $y$, and $z$ is generated to support information from all three axes. This strategy is addressed in related works [26], and it is essential to highlight that using images generated from individual axes does not take advantage of all the motion information [27]. Since the heart rate data consists of a single value (in bpm), the images are generated in grayscale (without performing RGB composition).

## Transforming time series into visual representations

Visual representations of time series, including Gramian Angular Field (GAF), Markov Transition Field (MTF), Recurrence Plots (RP) and spectrograms, provide practical advantages for analyzing one-dimensional data. Transforming time series into images captures complex features, such as temporal patterns, state transition dynamics, and recurrence structures. These methods allow data exploration in high-dimensional spaces, facilitating the extraction of robust features that are difficult to identify in one-dimensional formats. Additionally, this

approach supports advanced deep learning techniques like CNNs, which are optimized for detecting visual patterns and can automatically learn features without requiring manual segmentation or domain-specific expertise. This improves classification accuracy while reducing preprocessing requirements, making the process more efficient and scalable for various applications [28,29].

Time series were transformed into images using four types of representations: RP, GAF, MTF, and spectrograms. The details of each representation are presented in this subsection. Some examples of images obtained from accelerometer and heart rate data can be seen in Fig 4 and Fig 5, respectively.

**Recurrence plots (RP)** RP representations were proposed by Eckmann et al. [30] for non-linear analysis of time series data. These representations enable the visualization and acquisition of information about recurrent behavior in time series. The RP is an $N \times N$ matrix of points, where $N$ is the number of states, and a recurrence occurs when a trajectory revisits the same neighborhood in phase space as at some previous time.

The recurrence matrix $R$ can be described by Eq (1), where $\varepsilon$ is the recurrence threshold; $d(x[i], x[j])$ represents the distance between the values corresponding to time $i$ and $j$ in the time series $x$. The values of the time series are normalized before the transformation to RP.

$$R[i,j] = \begin{cases} 1, \text{ if } d(x[i], x[j]) \leq \varepsilon \\ 0, \text{ otherwise} \end{cases} \tag{1}$$

The traditional method of constructing RP can binarize the resulting matrix using different values for $\varepsilon$, some of which were explored in a previous work [27].

In addition to Eq (1), there are different variations of RP [31]. In this paper, the non-thresholded RP approach [32] (Eq (2)) was used, which maps the distances between pairs of points in a time series to a grayscale, providing a more compact visualization. For this,

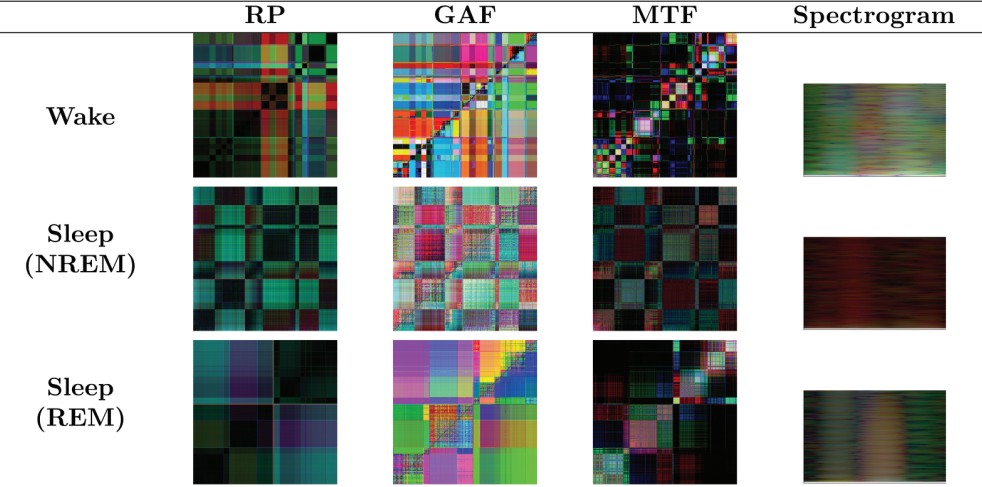

**Fig 4. Accelerometer data images.** Images generated from the accelerometer data for each type of representation and class. RGB images combine the x, y, and z axes to utilize all motion information. It is possible to observe visual differences between the classes, indicating that the visual representations capture specific motion patterns associated with each sleep stage.

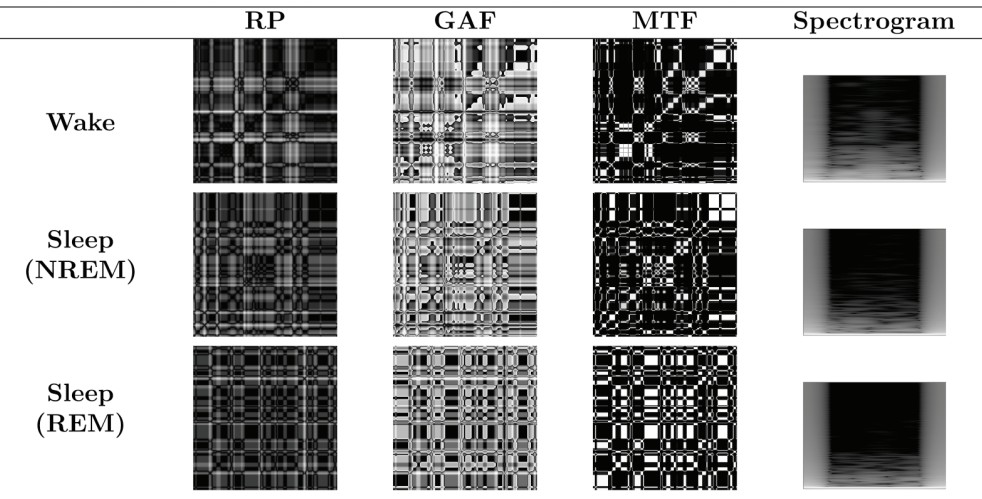

**Fig 5. Heart rate data images.** Images generated from the heart rate data for each type of representation and class. Grayscale images are used since heart rate data consists of a single value (in bpm). It is possible to observe visual differences between the classes, indicating that the visual representations capture specific heart rate patterns associated with each sleep stage.

the calculated distances are normalized. Then, the values are inverted so that smaller distances correspond to darker shades in the grayscale, indicating greater proximity or similarity between the analyzed points:

$$P[i,j] = \left\lfloor 255 \times \frac{d(x[i],x[j]) - d_{\min}}{d_{\max} - d_{\min}} \right\rfloor, \tag{2}$$

where:

- $d_{\min}$ is the minimum value in the distance matrix $d(x[i], x[j])$;
- $d_{\max}$ is the maximum value in the distance matrix $d(x[i], x[j])$;
- $\lfloor \cdot \rfloor$ represents the floor function, which rounds down to the nearest integer.

This equation normalizes the distances to the range of 0 to 255, transforming the distance matrix $d(x[i], x[j])$ into a pixel matrix $P[i,j]$ for visualization as a grayscale image.

**Gramian angular field (GAF)** The GAF representation [19] converts time series values into angles and uses these angles to generate a matrix that captures temporal relationships.

Given a time series $X = \{x_1, x_2, \ldots, x_N\}$, the transformation to GAF involves normalizing $X$ so that all its values are within the range $[-1, 1]$. Then, the angular transformation converts each normalized value $\tilde{x}_i$ into an angle $\phi_i$ using the arccosine function:

$$\phi_i = \arccos(\tilde{x}_i). \tag{3}$$

There are two main variants of GAF: Gramian Angular Summation Field (GASF) and Gramian Angular Difference Field (GADF). In this paper, the GADF representation, described in Eq (4), was used because it is the most suitable for highlighting trend changes in time series, as GADF emphasizes the angular differences between consecutive time

points [28]:

$$G_{i,j}^{(\text{GADF})} = \sin(\phi_i - \phi_j). \tag{4}$$

This matrix encodes the angular relationships between the time series points. It can be visualized as images in which patterns and structures can be identified.

**Markov transition field (MTF)**  Each element of the MTF matrix [19] reflects the probability of transitioning between two states at different times in the time series, capturing the temporal dynamics of the series in a two-dimensional representation.

Given a time series $X = \{x_1, x_2, \ldots, x_N\}$, the construction of the MTF is carried out by discretizing $X$ into $Q$ quantiles. This paper used $Q = 8$, as described in a related work [33]. Each value $x_i$ in $X$ is assigned to a corresponding quantile, resulting in a discretized series $\tilde{X} = \{\tilde{x}_1, \tilde{x}_2, \ldots, \tilde{x}_N\}$, where $\tilde{x}_i$ represents the quantile to which $x_i$ belongs.

The Markov transition matrix $W$, of dimension $Q \times Q$, represents $w_{ij}$ as the frequency with which a point in quantile $q_j$ is followed by a point in quantile $q_i$ ($q_j \rightarrow q_i$). After normalization, where $\sum_{j=1}^{Q} w_{ij} = 1$ for each $i$, $W$ becomes the Markov transition matrix. The matrix $W$ before normalization is a count transition matrix and can be visualized as follows:

$$W = \begin{bmatrix} w_{11} & w_{12} & \cdots & w_{1Q} \\ w_{21} & w_{22} & \cdots & w_{2Q} \\ \vdots & \vdots & \ddots & \vdots \\ w_{Q1} & w_{Q2} & \cdots & w_{QQ} \end{bmatrix}, \tag{5}$$

where $w_{ij}$ is the count of transition occurrences in the time series.

The MTF is a $Q \times Q$ matrix, and in its construction, each element $M_{ij}$ denotes the probability of transitioning from quantile $q_i$ to quantile $q_j$, considering the temporal positions in the series. MTF thus encodes the multi-scale transition probabilities of the time series. For instance, $M_{ij}$ with $|i - j| = k$ represents the probability of transition between points with a temporal interval $k$. The main diagonal $M_{ii}$ captures the self-transition probabilities.

**Spectrograms**  This transformation is performed by applying the Short-Time Fourier Transform (STFT), which essentially decomposes the signal into its component frequencies at different time instances, allowing visualization of how the signal's frequency spectrum varies over time [34].

Given a time series $x(t)$, the STFT is defined as:

$$STFT\{x(n)\}(m, k) = \sum_{n=-\infty}^{\infty} x(n) w(n - m) e^{-j2\pi k(n-m)/N}, \tag{6}$$

where

- $x(n)$ is the value of the time series at time $n$;
- $w(n-m)$ is the window function applied to the signal, shifted by the frame index $m$;
- $N$ is the total number of points used in the Fourier transform, influencing frequency resolution;
- $e^{-j2\pi k(n-m)/N}$ represents the Fourier transform basis;
- $m$ indicates the current frame position;
- $k$ is the frequency index.

The Hann window function is defined as

$$w(n) = \begin{cases} 0.5 \times \left[ 1 - \cos\left( \dfrac{2\pi n}{M-1} \right) \right], & \text{if} \quad 0 \leq n \leq M-1 \\ 0, & \text{otherwise.} \end{cases} \tag{7}$$

The time series is divided into smaller, overlapping time segments. The Fourier transform assumes the signal is periodic, which is not valid for most real signals. Therefore, the window limits the signal to a finite time interval, allowing for local frequency analysis. The Fourier transform is then applied to each time segment, transforming the data from the time domain to the frequency domain. This provides the amplitude and phase of the frequencies present in each time segment.

The results of the Fourier transform for each segment are organized into a matrix, where one dimension represents time (the time segments), and the other represents frequency. The matrix values represent the frequencies' magnitude in each time segment.

The squared magnitude of the STFT is often used to construct the spectrogram of the signal, which is a visual representation of the intensity of frequencies as a function of time:

$$\text{Spectrogram}(x(n)) = |STFT\{x(n)\}(m,k)|^2. \tag{8}$$

In the spectrogram, the $x$ axis represents time, the $y$ axis represents frequency, and the intensity of grayscale at a specific point represents the magnitude of the frequency at that time point.

## Patches

The technique of dividing an image into sub-images, or patches, is a common approach in image visualization and machine learning applications [35,36], primarily aimed at improving focus on local details of the image. This allows the model to learn finer features that might be missed when observing the original image.

As Fig 6 depicts, each image generated by different representations (600×600 pixels or other dimensions, in the case of spectrograms) is divided into nine patches, and each patch (224×224 pixels, with approximately 16% overlap between patches to avoid losing local information) is treated as an independent input for the training process. After training, the patches can be regrouped to make predictions about the entire image. A model is trained on each patch to predict the class of the corresponding patch region. Then, these predictions are combined to obtain the complete segmented image prediction.

## Ensembles

Ensemble techniques combine the predictions of multiple models to improve the robustness and accuracy of the final prediction, aiming to leverage individual decisions and mitigate the drawbacks of each model. In this context, an ensemble uses the predictions of models trained with:

- original images of accelerometer data + original images of heart rate data: two models (one for each type of sensor);
- patches of accelerometer data: nine models (one for each patch);
- patches of heart rate data: nine models (one for each patch);
- patches of accelerometer data + patches of heart rate data: 18 models (nine models for accelerometer data patches + nine models for heart rate data patches).

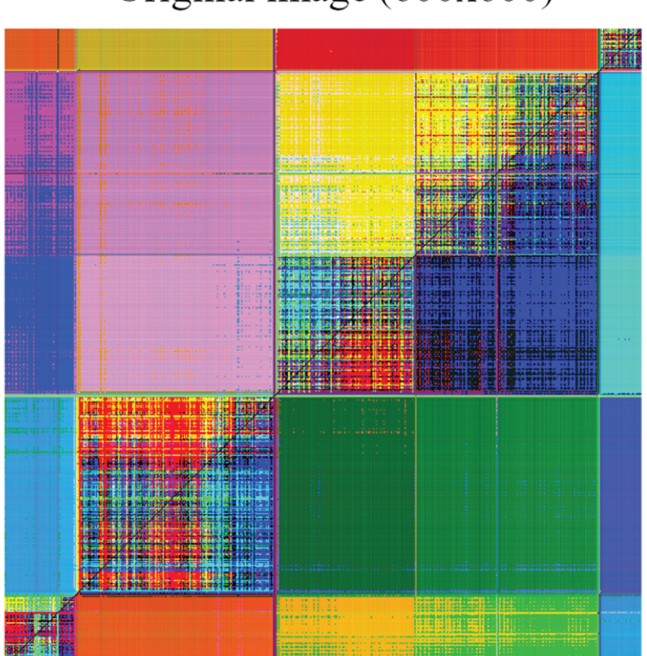 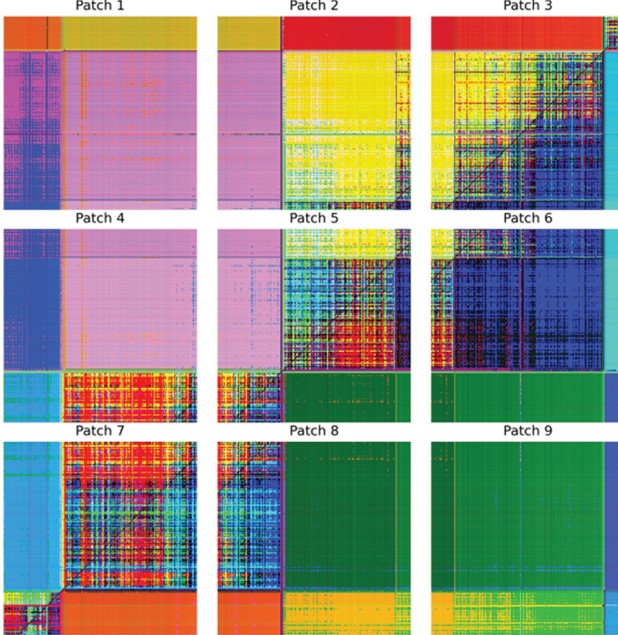

**Fig 6. "Wake" images generated with GAF and accelerometer data.** Example of an original image (600×600) and examples of patches (224×224 each).

These techniques are helpful in complex problems where more than a single model may be required to capture all the nuances of the data [37]. The ensembles of simple averaging, weighted averaging, simple network, and deep features are described below.

**Simple averaging** In the simple averaging ensemble, the predictions of each classifier are combined by calculating the arithmetic mean of the predictions. For the binary classification problem, this means calculating the mean of the predicted probabilities for the classes "Sleep" and "Wake" by the classifiers. For the three-class classification, the mean of the predictions for each class is calculated individually. This type of ensemble was applied to combine the predictions of models trained with:

- original images from accelerometer data + original images from heart rate data;
- patches from accelerometer data;
- patches from heart rate data; and
- patches from accelerometer data + patches from heart rate data.

**Weighted averaging** Weighted averaging is a variation of the simple averaging method, where each classifier's prediction contributes a different weight to the final prediction. The weights are usually assigned based on each classifier's performance. We generated 1000 random sets of $n$ weights to find the best combination in each case. This type of ensemble was applied to combine the predictions of models trained with:

- original images from accelerometer data + original images from heart rate data ($n = 2$ number of sensors);
- patches from accelerometer data ($n = 9$ number of patches);

- patches from heart rate data ($n$ = 9 number of patches); and
- patches from accelerometer data + patches from heart rate data ($n$ = 9 × 2 = number of patches × number of data types).

**Simple network**  In this method, the classifiers' predictions are input to a simple neural network, which learns the best way to combine these predictions. The predicted probabilities for the classes of interest by a classifier trained with patches are the inputs to this network, providing the final prediction. To maintain only one simple network, this ensemble was applied only with the predictions of models trained with:

- patches from accelerometer data; and
- patches from heart rate data.

Using models trained only with patches, multiple predictions contribute to a more informative input, unlike using accelerometer and heart rate data, where only two models would contribute. This makes a simple network less advantageous for combining only two predictions due to the additional effort not justified by the problem's complexity.

**Deep features**  In this type of ensemble, feature vectors are extracted from the deep layers of each classifier and combined to be used as input for a final model. This final model is trained to make the final prediction using these combined features, leveraging the data representations provided by the different classifiers. This type of ensemble was applied to combine the predictions of models trained with original images from accelerometer data + original images from heart rate data.

Although the diversity of information provided by the various deep feature vectors from patches can be informative, it also introduces significant complexity to the modeling process. This complexity manifests in the data dimensions to be processed. Working with only two deep feature vectors generated from accelerometer and heart rate data simplifies the modeling process.

## Training and validation

For training, we employed transfer learning using the EfficientNet-B0 model, a specific variant of the EfficientNet family [38], pre-trained on the extensive ImageNet dataset [39]. The strategy includes freezing some of the initial layers of these networks to preserve the learned generic features while the deeper layers are adapted to the specific dataset. This adaptation was carried out by adding a dense network at the end of the architecture, a process known as fine-tuning, allowing fine adjustments of the network parameters to fit the classes of interest better. EfficientNet-B0 was chosen due to its balance between high accuracy and computational efficiency, making it well-suited for tasks involving image classification. Additionally, other architectures, such as ResNet network [40], were also tested. However, EfficientNet-B0 consistently provided better performance in terms of accuracy and training time for the specific dataset used in this study.

The decision to freeze 90% of the initial layers of the EfficientNet-B0 network was based on a layer-freezing experiment. Different percentages of layers were frozen, ranging from 50% to 100%, and their impact on model performance was analyzed. Freezing 90% of the layers yielded the best trade-off, preserving general features while allowing the deeper layers to adapt to the dataset. Through these layer-freezing tests, it was established to use 90% of the first layers of the EfficientNet-B0 network frozen. Additionally, a dense layer of size 512 with 50% dropout and a dense layer of size 256 with 20% dropout were added.

The *k-fold* cross-validation technique was applied, with $k = 5$, dividing the dataset into five distinct partitions to ensure that each sample was not used for both training and validation to evaluate the robustness and generalization of the models. We recall that data from the same subject were not simultaneously used for training and validation. Unlike stratified cross-validation, which preserves label distributions in each fold, we opted for a random split to maintain the natural variability of sleep stage transitions in real-world sleep patterns.

To determine whether the random split introduced significant discrepancies, we analyzed the class distribution in each split. Table 2 presents the percentage of Wake, NREM, and REM samples in the training and validation sets for sleep stage classification. For sleep/wake classification, where Sleep includes both NREM and REM, the distribution for Wake remains the same, while Sleep corresponds to the sum of NREM and REM. The results indicate that the overall distribution remains stable across splits, particularly in the training data, ensuring a balanced representation during model learning. While the validation distribution shows some variability, particularly in Split 1 and Split 2 for Wake and Split 5 for REM, this reflects real-world sleep data, where sleep stages are inherently imbalanced across different nights and individuals. Since the training data maintains a consistent distribution and the model is evaluated across multiple folds, the impact of these variations is minimized. Thus, the use of a random split does not introduce substantial bias or compromise the reliability of the results, as it allows the model to be tested under conditions that resemble real sleep patterns.

Recognizing the challenge posed by class imbalance in the dataset, the class weighting technique was applied, where weights are assigned to each class inversely proportional to their frequency in the dataset. This approach ensures that minority classes contribute more significantly to the loss function during training, preventing the model from being biased toward the majority class. By adjusting the importance of each class in this way, class weighting helps mitigate the imbalance effect, leading to a more equitable and representative training process.

The class weights $w_c$ were calculated as the inverse of the class frequencies, normalized by the total number of samples:

$$w_c = \frac{N}{C \times n_c} \tag{9}$$

where $N$ is the total number of samples, $C$ is the number of classes, $n_c$ is the number of samples in class $c$. For binary cross-entropy, which is used for sleep/wake classification, the weighted loss $\mathcal{L}$ is computed as:

$$\mathcal{L} = -\frac{1}{N} \sum_{i=1}^{N} \left[ w_{\text{pos}} \times y_i \times \log(p_i) + w_{\text{neg}} \times (1 - y_i) \times \log(1 - p_i) \right] \tag{10}$$

**Table 2**. **Class distribution per split.** Percentage of Wake, NREM, and REM samples in the training and validation sets across the five splits of the cross-validation. The data indicate that the class distribution remains stable across splits, suggesting that the random split does not introduce substantial bias.

| Split | Train | | | | Validation | | | |
|---|---|---|---|---|---|---|---|---|
| | Folds | Wake (%) | NREM (%) | REM (%) | Folds | Wake (%) | NREM (%) | REM (%) |
| 1 | 2, 3, 4, 5 | 8.0 | 69.8 | 22.2 | 1 | 10.5 | 69.6 | 19.9 |
| 2 | 1, 3, 4, 5 | 9.3 | 69.8 | 20.9 | 2 | 5.3 | 69.6 | 25.1 |
| 3 | 1, 2, 4, 5 | 8.4 | 70.3 | 21.3 | 3 | 9.1 | 67.3 | 23.6 |
| 4 | 1, 2, 3, 5 | 8.6 | 70.1 | 21.3 | 4 | 8.3 | 68.3 | 23.4 |
| 5 | 1, 2, 3, 4 | 8.3 | 68.7 | 23.0 | 5 | 9.4 | 74.0 | 16.6 |

where $y_i$ the true label (0 or 1) for sample $i$, $p_i$ is the predicted probability for the positive class, $w_{pos}$ and $w_{neg}$ are the weights for the positive and negative classes, respectively. For categorical cross-entropy, which is used for sleep stages classification (three classes), the weighted loss $\mathcal{L}$ is computed as:

$$\mathcal{L} = -\frac{1}{N} \sum_{i=1}^{N} \sum_{c=1}^{C} w_c \times y_{i,c} \times \log(p_{i,c}) \tag{11}$$

where $y_{i,c}$ is a binary indicator (0 or 1) for whether class $c$ is the correct classification for sample $i$, $p_{i,c}$ is the predicted probability for class $c$, $w_c$ is the weight assigned to class $c$.

## Performance metrics and model evaluation

Although related works present accuracy as the main performance measure of the model, this paper uses balanced accuracy, given that the data contains imbalanced classes and that accuracy provides an optimistic estimate when a classifier is tested on an unbalanced dataset [41]. Eq (13) describes balanced accuracy, where $c$ is the number of classes. In Eq (12), $TP_n$ is the number of true positives of class $n$, and $FN_n$ is the number of false negatives of class $n$. Sensitivity and Cohen's kappa coefficient ($\kappa$) were also used as metrics. In this context, the evaluators are the PSG labels and the automatic sleep stages classification algorithm. Cohen's definition of $\kappa$ [42] is described in Eq (14).

The reported results correspond to the average obtained from 5-fold cross-validation. The standard deviation of balanced accuracy across the folds is also presented to quantify performance variability.

$$\text{sensitivity}_n = \frac{TP_n}{TP_n + FN_n} \tag{12}$$

$$\text{balanced accuracy} = \frac{\sum_{n=1}^{n=c} \text{sensitivity}_n}{c} \tag{13}$$

$$\kappa = \frac{\%\text{observed agreement} - \%\text{agreement by chance}}{1 - \%\text{agreement by chance}} \tag{14}$$

## Results and discussion

Here, we present the main experiments and results obtained with the Sleep Accel database [25] for sleep/wake and sleep stages classifications, along with the discussions. Subsequently, a comparison of the visual representation with other data representations typically used in related works is conducted.

## Sleep/wake classification

This subsection presents the balanced accuracies, $\kappa$ coefficients, and confusion matrices for sleep/wake classification. Additionally, we analyze a subject's night's sleep within this context.

**Balanced accuracy and Cohen's $\kappa$** Table 3 presents the balanced accuracies obtained with each sleep/wake classification representation. The complete table, which shows the results for individual patches, is available in S1 Table.

Compared to the heart rate data, the accelerometer data presents better results with all representations, both for the original images and for the individual patches and the ensembles of individual patches, for this scenario.

Observing the results with the ensembles that combine the different types of data (ACC and HR), it is noted that in no case do these present better-balanced accuracies compared to

**Table 3. Balanced accuracies obtained with each representation for sleep/wake classification.** Accelerometer data consistently outperformed heart rate data in all scenarios, with the GAF achieving the highest balanced accuracy (82.36% ± 3.24%) when using patch ensembles. Patch-based ensembles significantly improved balanced accuracy compared to original images.

| Network | Config. | RP ACC | RP HR | GAF ACC | GAF HR | MTF ACC | MTF HR | Spectrograms ACC | Spectrograms HR |
|---|---|---|---|---|---|---|---|---|---|
| Eff.Net | Original | 76.62 | 69.91 | 79.63 | 69.32 | 77.44 | 64.88 | 78.34 | 61.22 |
| ACC + HR Ensembles | Simple Average | 71.38 | | 69.04 | | 71.27 | | 75.94 | |
| | Weighted Average | 76.54 | | 77.57 | | 74.95 | | 78.19 | |
| | Deep Features | 75.59 | | 77.61 | | 75.43 | | 77.12 | |
| Ensembles of Patches | Simple Average | 80.39 | 71.39 | 82.36 | 73.71 | 80.03 | 70.21 | 79.11 | 54.90 |
| | Weighted Average | 80.38 | 71.28 | 82.04 | 72.15 | 80.20 | 68.78 | 79.01 | 54.91 |
| | Simple Network | 80.26 | 70.81 | 81.54 | 72.87 | 79.85 | 69.41 | 79.03 | 53.30 |
| P. ACC + P. HR Ensembles | Simple Average | 76.14 | | 77.25 | | 78.28 | | 77.78 | |
| | Weighted Average | 79.28 | | 81.44 | | 80.32 | | 78.64 | |

Note: The underlined values represent the highest balanced accuracies for the corresponding data/network configurations.

those obtained with the accelerometer data. In other words, the results obtained with heart rate data combined with accelerometer data do not contribute to a gain in balanced accuracy in this scenario. In turn, the results obtained with the ensembles of individual patches, both for the accelerometer data and the heart rate data, show a significant gain compared to the results obtained with the original images (up to 3.7 percentage points for RP, up to 2.7 for GAF, up to 2.9 for MTF, and up to 0.7 for Spectrograms).

Comparing the different representations, GAF achieved 82% balanced accuracy with the ensemble of patches through simple averaging and accelerometer data, indicating the best sleep/wake classification result. The RP and MTF representations show balanced accuracies greater than 80% with ensemble patches and accelerometer data (except MTF and an ensemble of patches with the simple network). With GAF and MTF, the ensemble of ACC patches + HR patches also shows results greater than 80%, and the Spectrograms exceed 78%.

The standard deviations obtained from the balanced accuracies for two-stage sleep classification are up to 3% for RP, up to 4% for GAF, up to 6% for MTF, and up to 4% for Spectrograms.

The best $\kappa$ coefficients are concentrated in the ACC + HR ensemble results with deep features, exceeding 0.4 for the GAF representation (moderate agreement). For most configurations (except MTF patches 1 and 3), the $\kappa$ obtained with accelerometer data exceeds 0.2, indicating fair agreement. Whereas with heart rate data, $\kappa$ exceeds 0.2 only with the application of ensembles (and with GAF patch 6) and is lower with Spectrograms. This again highlights the importance of accelerometer data for sleep/wake classification.

**Confusion matrices** Fig 7-10 show the confusion matrices for sleep/wake classification using RP, GAF, MTF and Spectrogram representations, respectively. These are generated with original accelerometer data, heart rate data and the best accelerometer + heart rate ensembles. Additionally, the figures include the confusion matrices for the best ensembles of accelerometer data patches heart rate data patches and accelerometer data patches + heart rate data patches.

The confusion matrices for the RP representation (Fig 7) reveal that the ensemble combining the original accelerometer and heart rate data achieves higher sensitivity than other approaches. Comparing confusion matrices from original data versus patches highlights an improvement in classifying the "Wake" stage. For accelerometer data and heart rate data, the use of patches increased the correct classification of "Wake." Sleep/wake classification is

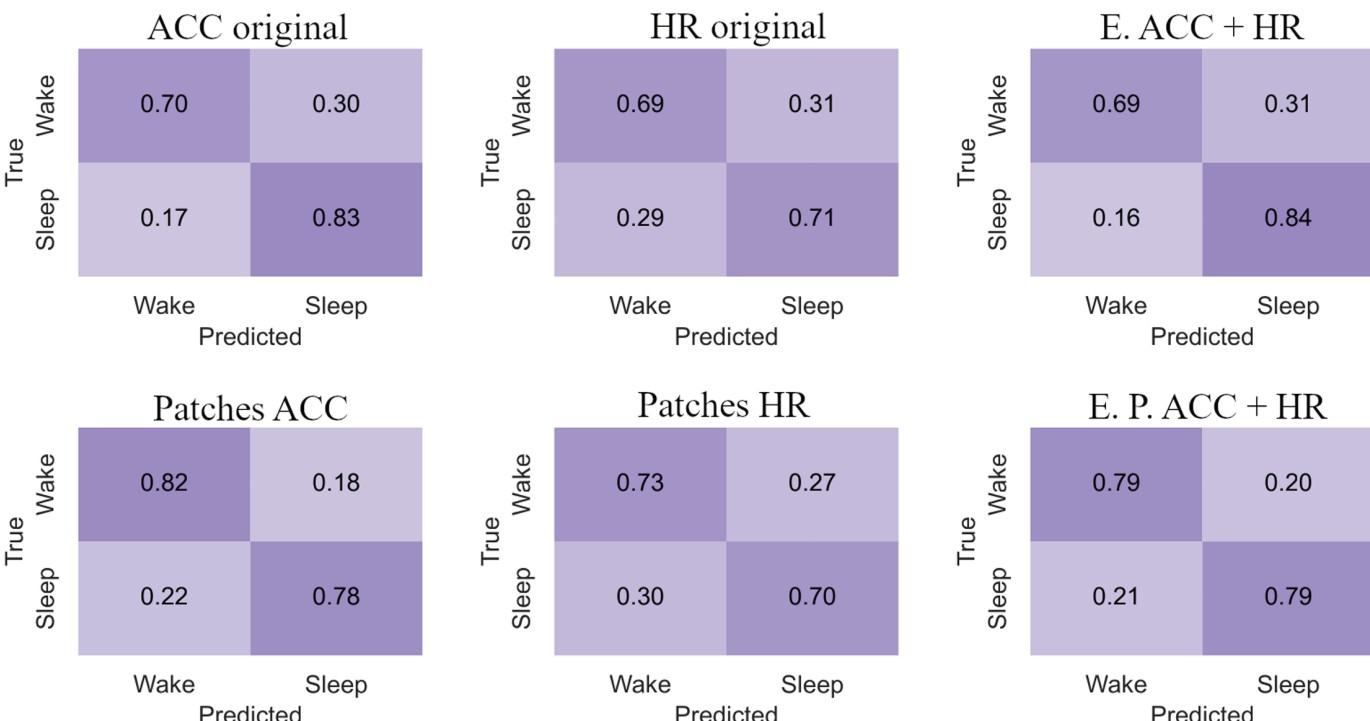

**Fig 7. RP confusion matrices for sleep/wake classification.** The highest balanced accuracy (80.39% ± 2.43%) was achieved with accelerometer data and the ensemble of patches, while the highest sensitivity (84%) was observed with the ensemble combining accelerometer and heart rate. Ensemble combining the original accelerometer and heart rate data achieves higher sensitivity than other approaches. Comparing confusion matrices from original data versus patches highlights an improvement in classifying the "Wake" stage. For accelerometer data and heart rate data, the use of patches increased the correct classification of "Wake". Sleep/wake classification is more balanced when using accelerometer data, and this balance is further enhanced in the ensemble combining accelerometer and heart rate patches.

more balanced when using accelerometer data, and this balance is further enhanced in the ensemble combining accelerometer and heart rate patches.

For the GAF representation (Fig 8), sleep/wake classification using original heart rate data is more balanced than with original accelerometer data, where both classes are confused to a similar extent. Classification with original accelerometer data tends to overestimate "Sleep." However, by improving predictions for "Wake" using patches, the accelerometer-based classification becomes more balanced. The ensemble of accelerometer and heart rate patches achieves the highest sensitivity for this representation at 85%.

In the MTF representation (Fig 9), using original accelerometer and heart rate data, "Sleep" is classified more accurately than "Wake." This pattern is reflected in the ensemble of original data. As observed in the RP and GAF representations, the use of patches for accelerometer data leads to a more balanced classification. Consequently, the ensemble of accelerometer and heart rate patches is also more balanced than the ensemble of original data, achieving a sensitivity of 87%.

The Spectrogram representation (Fig 10) shows balanced classifications for both original accelerometer and heart rate data. However, the ensemble of these original data better classifies "Sleep" stages. Using patches for accelerometer data improves the classification of "Wake". Both ensembles show slightly better performance in classifying "Sleep" stages, achieving the highest sensitivity for this representation at 80%.

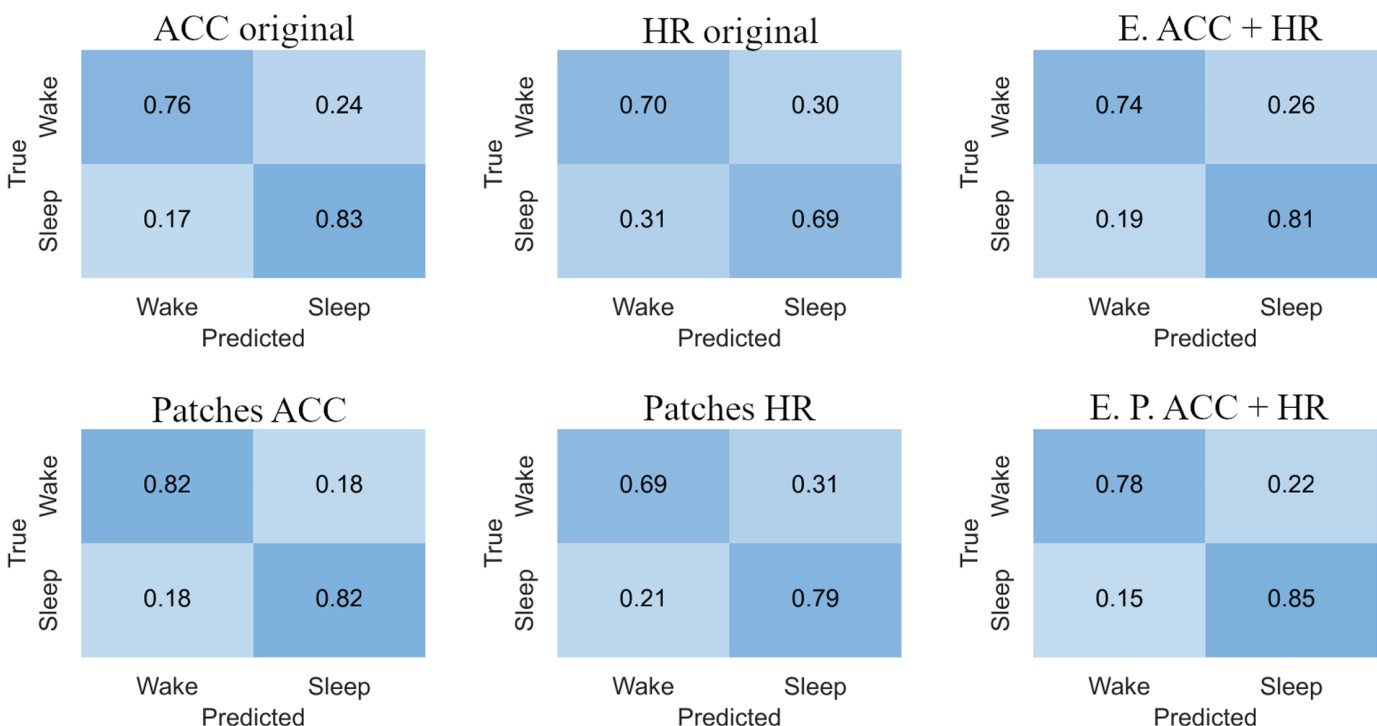

**Fig 8. GAF confusion matrices for sleep/wake classification.** The highest balanced accuracy (82.36% ± 3.24%) was achieved with accelerometer data and the ensemble of patches, while the highest sensitivity (85%) was observed with the ensemble combining accelerometer and heart rate patches. Sleep/wake classification using original heart rate data is more balanced than with original accelerometer data, where both classes are confused to a similar extent. Classification with original accelerometer data tends to overestimate "Sleep". However, by improving predictions for "Wake" using patches, the accelerometer-based classification becomes more balanced.

**Analysis of a night of sleep** Analyzing an entire night of sleep provides a comprehensive view of the performance of the model in a real world scenario, where transitions between sleep and wake states occur naturally. This type of analysis is particularly useful for evaluating the consistency of predictions over extended periods and identifying potential limitations in the model's ability to capture subtle transitions or irregularities.

Fig 11 illustrates the predictions of a night's sleep for the same subject for sleep/wake classification using original accelerometer data and the ensemble of patches of accelerometer data. With these figures, it is possible to compare the best night's sleep obtained for sleep/wake classification (ensemble of accelerometer data patches) obtained through GAF with the corresponding night's sleep using original data. It can be observed, primarily, the decrease in "Sleep" prediction errors when the true class is "Wake" using the ensemble of patches. This fact was also illustrated in the corresponding confusion matrix (Fig 8).

With the original data from the accelerometer, errors from "Sleep" to "Wake" occur more frequently in the early part of the night and, around 6 hours of sleep, this type of error occurs occasionally. The "Wake" errors for "Sleep" also occur more often during this initial phase. Using the ensemble of patches of accelerometer data, some "Sleep" errors for "Wake" occur only at the beginning of the night, while the "Wake" errors for "Sleep" occur more frequently around 2 and 6 hours of sleep.

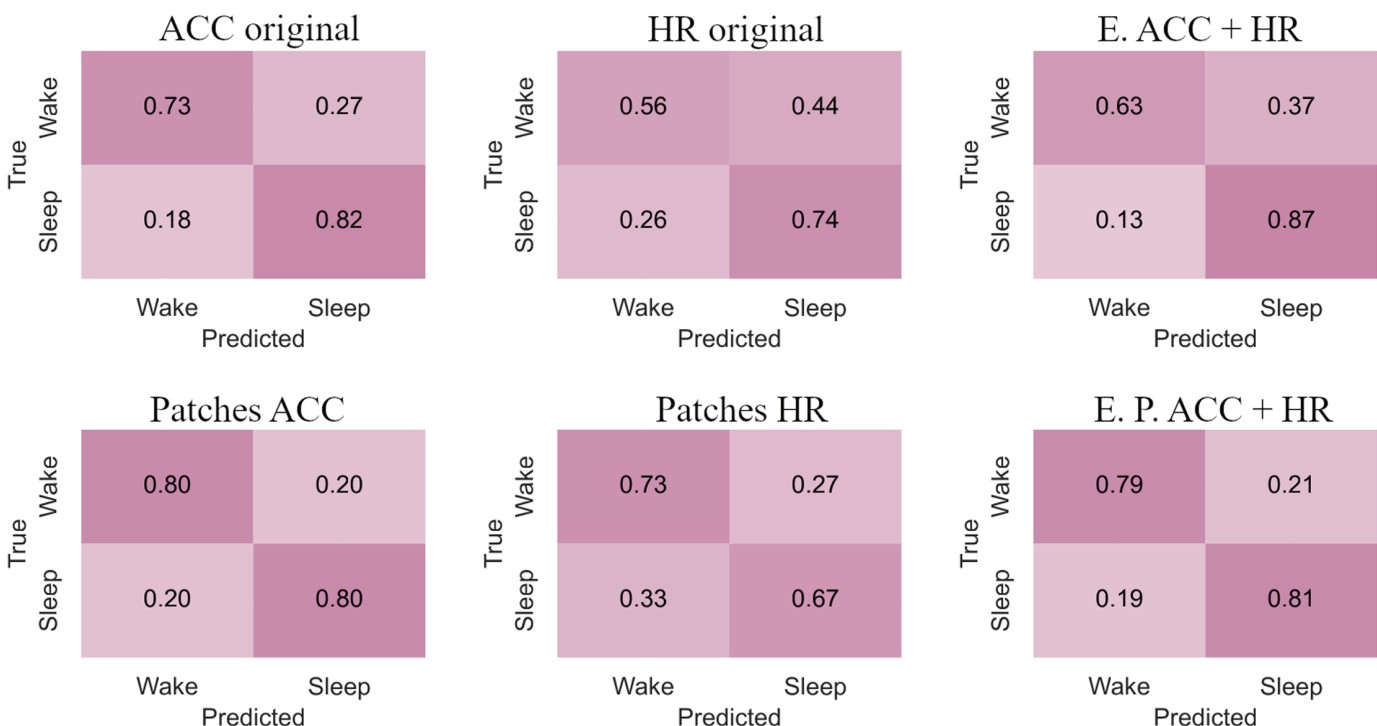

**Fig 9. MTF confusion matrices for sleep/wake classification.** The highest balanced accuracy (80.32% ± 3.43%) was achieved with the ensemble combining accelerometer and heart rate patches, while the highest sensitivity (87%) was observed with the ensemble combining accelerometer and heart rate. Using original accelerometer and heart rate data, "Sleep" is classified more accurately than "Wake". This pattern is reflected in the ensemble of original data. As observed in the RP and GAF representations, the use of patches for accelerometer data leads to a more balanced classification.

## Sleep stages classification

This subsection presents the balanced accuracies, $\kappa$ coefficients, and confusion matrices obtained for sleep stages classification, along with an analysis of a subject's night of sleep in this scenario.

**Balanced accuracy and Cohen's $\kappa$** Table 4 presents the balanced accuracies obtained with each representation for sleep stages classification. The complete table, which shows the results for individual patches, is available in S2 Table.

Analyzing Table 4, it is noted that, in contrast to the results obtained for sleep/wake classification, the balanced accuracies displayed with heart rate data are higher than those with accelerometer data in many cases (except for the Spectrogram representation). To combine ACC + HR, the technique that presented the best-balanced accuracies was the ensemble of ACC patches + HR patches through weighted averaging, obtaining balanced accuracy above 60% for most representations (except Spectrograms).

Using Spectrograms, the ACC + HR ensemble, through weighted averaging with the original configuration data, showed an advantage over the individual original accelerometer and heart rate data. In contrast, the other representations did not improve balanced accuracy when applying this type of ensemble. On the other hand, the ensemble of ACC patches + HR patches with weighted averaging presented better balanced accuracies than the ensemble of accelerometer data patches for most representations. This again highlights the importance of heart rate data patches for this scenario.

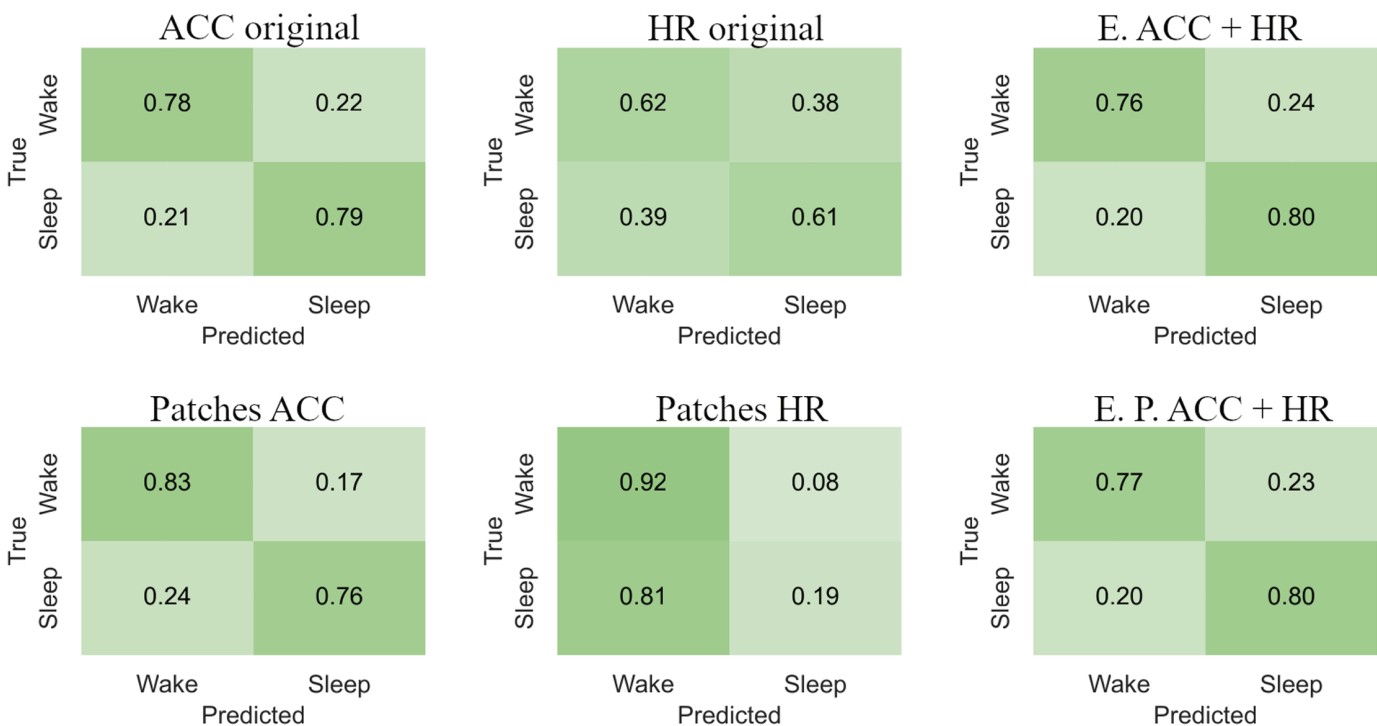

**Fig 10. Spectrograms confusion matrices for sleep/wake classification.** The highest balanced accuracy (79.11% ± 3.93%) was achieved with accelerometer data and the ensemble of patches, while the highest sensitivity (80%) was observed with both ensembles combining accelerometer and heart rate. Spectrogram representation shows balanced classifications for both original accelerometer and heart rate data. However, the ensemble of these original data better classifies "Sleep" stages. Using patches for accelerometer data improves the classification of "Wake".

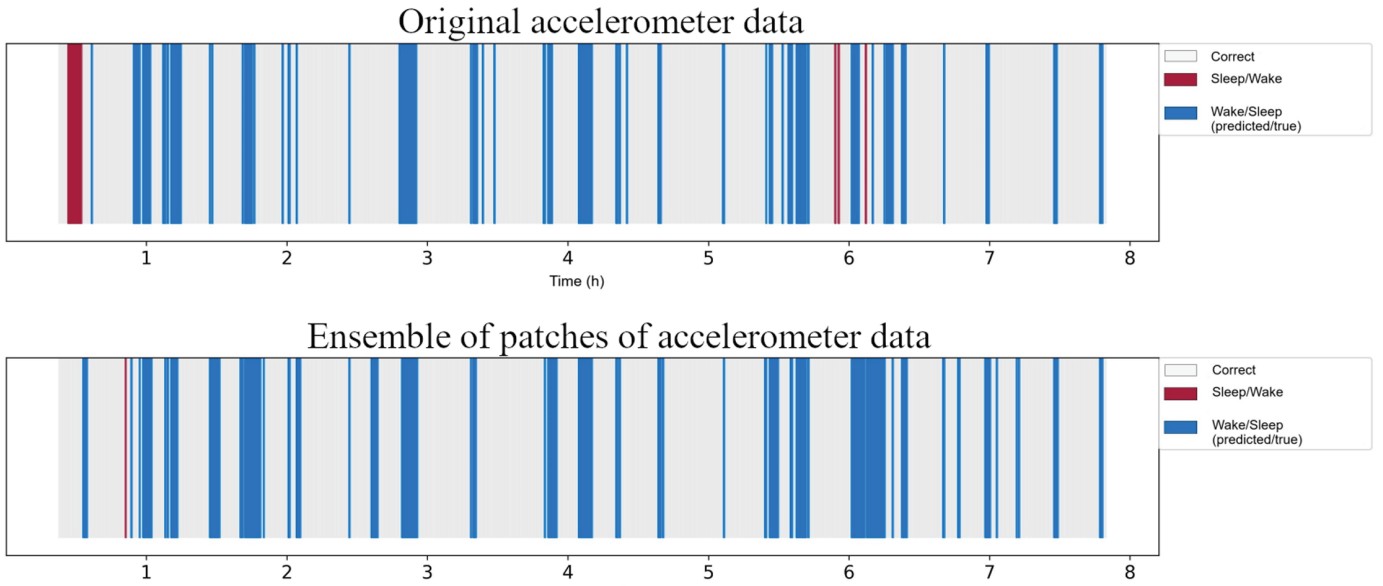

**Fig 11. Sleep/wake classification over a night of sleep for a subject using the GAF representation.** Original data shows more "Sleep" errors for "Wake" at the beginning and around 6 hours, and frequent "Wake" errors for "Sleep" early on. Ensemble of patches reduces "Sleep" errors for "Wake", with most "Wake" errors for "Sleep" around 2 and 6 hours.

**Table 4. Balanced accuracies obtained with each representation for sleep stages classification.** Heart rate data often outperformed accelerometer data in balanced accuracies (except for the Spectrogram), with the GAF achieving the highest balanced accuracy (62.18% ± 0.95%) when using patch ensemble. Patch-based ensembles significantly improved balanced accuracy compared to original images.

| Network | Config. | RP | | GAF | | MTF | | Spectrograms | |
|---|---|---|---|---|---|---|---|---|---|
| | | ACC | HR | ACC | HR | ACC | HR | ACC | HR |
| Eff.Net | Original | 55.73 | 57.85 | 57.68 | 57.09 | 55.00 | 53.35 | 55.96 | 40.32 |
| | Simple Average | 46.01 | | 46.14 | | 46.35 | | 52.41 | |
| | Weighted Average | 50.24 | | 50.10 | | 50.37 | | 56.00 | |
| ACC + HR Ensembles | Deep Features | 53.60 | | 53.39 | | 54.82 | | 51.01 | |
| | Simple Average | 59.19 | 61.87 | 60.66 | 62.18 | 58.38 | 57.81 | 55.53 | 39.50 |
| | Weighted Average | 59.41 | 61.46 | 60.16 | 61.57 | 58.30 | 57.34 | 57.36 | 39.28 |
| Ensembles of Patches | Simple Network | 49.96 | 52.44 | 51.19 | 51.25 | 48.29 | 51.29 | 48.69 | 39.33 |
| | Simple Average | 49.85 | | 49.98 | | 50.95 | | 53.10 | |
| P. ACC + P. HR Ensembles | Weighted Average | 61.48 | | 61.17 | | 60.97 | | 55.35 | |

Note: The underlined values represent the highest balanced accuracies for the corresponding data/network configurations.

As with sleep/wake classification, the best results of all representations involve ensembles of patches, exceeding 62% with GAF, 61% with RP, 60% with MTF, and 57% with Spectrograms. Comparing the best-balanced accuracies obtained and the original configuration of each representation, it is possible to observe gains of up to 4.0 percentage points for RP, 5.0 for GAF, 6.0 for MTF, and 1.4 for Spectrograms.

The standard deviations obtained from the balanced accuracies for sleep stages classification are up to 6% for RP, up to 3% for GAF, up to 2% for MTF, and up to 3% for Spectrograms.

As observed with the balanced accuracies, the best $\kappa$ were obtained with experiments involving ensembles of patches. Except for Spectrograms, the best $\kappa$ obtained for sleep stages classification were achieved with the ensemble of heart rate data patches (weighted averaging for RP and GAF and simple network for MTF), being $\kappa = 0.38$ for RP (fair agreement), $\kappa = 0.41$ for GAF (moderate agreement), and $\kappa = 0.32$ for MTF (fair agreement). Again, this indicates the relevance of this type of data for sleep stages classification.

**Confusion matrices** Figs 12-15 respectively show the confusion matrices generated for sleep stages classification using the RP, GAF, MTF and Spectrogram representations with the original accelerometer data, heart rate data and the best accelerometer + heart rate ensembles. The figures also include the confusion matrices of the best ensembles of accelerometer data patches, heart rate data patches and the best ensembles of accelerometer data patches + heart rate data patches.

The confusion matrices generated with RP (Fig 12) indicate that accelerometer data, including the ensemble results, achieved a higher number of correct classifications for "Wake." In contrast, matrices generated with heart rate data alone showed more accurate classifications of "NREM" and "REM". Additionally, with accelerometer data (both original and patches), the most frequent misclassification was labeling "NREM" as "REM." For heart rate data, the most common error was classifying "Wake" as "REM."

For the GAF representation (Fig 13), the confusion matrix with heart rate patches demonstrated an increase in correct classifications of "NREM" and "REM". The most common misclassifications with accelerometer data were labeling "NREM" as "REM" and "REM" as "NREM." Meanwhile, with heart rate data, the most frequent error was classifying "Wake" as "REM."

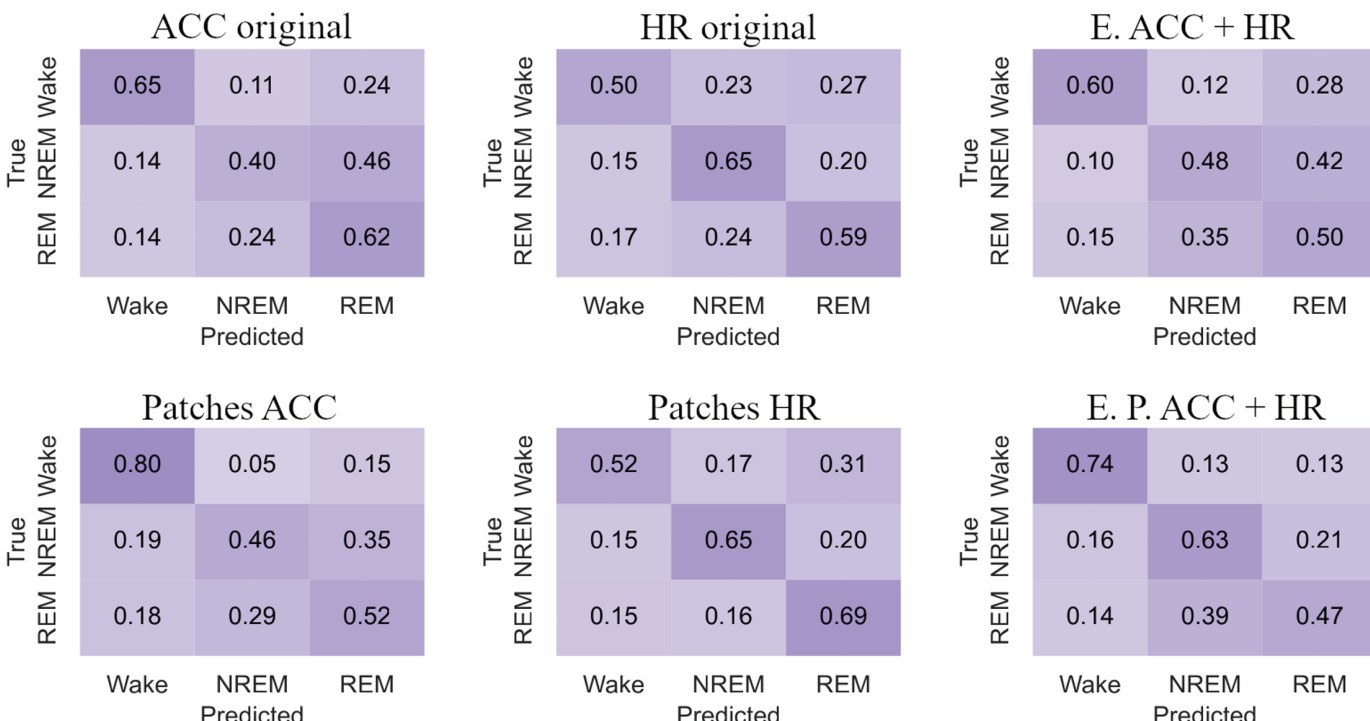

**Fig 12. RP confusion matrices for sleep stages classification.** The highest balanced accuracy (61.87% ± 1.67%) was achieved with heart rate data and the ensemble of patches. Accelerometer data, including the ensemble results, achieved a higher number of correct classifications for "Wake". In contrast, matrices generated with heart rate data alone showed more accurate classifications of "NREM" and "REM". Additionally, with accelerometer data (both original and patches), the most frequent misclassification was labeling "NREM" as "REM". For heart rate data, the most common error was classifying "Wake" as "REM".

The confusion matrices for the MTF representation (Fig 14) show that accelerometer data and both types of ensembles most frequently classified "Wake" correctly, similar to other representations. For heart rate data, the incorrect classification of "Wake" as "REM" observed with original data decreased with the use of patches, resulting in a more balanced classification.

With the Spectrogram representation (Fig 15), the confusion matrices suggest that heart rate data, both in its original and patched forms, resulted in fewer classifications of "NREM" and "REM," overestimating "Wake." However, the patched configuration increased the number of correct "REM" classifications while reducing incorrect "Wake" predictions. Both ensemble configurations improved correct classifications of "Wake" but showed increased confusion for "NREM" when the true class was "REM."

For accelerometer data, "NREM" is often misclassified as "REM," suggesting overlapping movement features. In contrast, heart rate data frequently misclassifies "Wake" as "REM," possibly due to pattern similarities. Patched configurations reduce these errors, improving classification balance. For example, "Wake" misclassified as "REM" decreases with heart rate patches, showing their effectiveness in refining feature representation. "Wake" is consistently the most accurately classified stage across all representations (RP, GAF, MTF, Spectrogram), reflecting its distinct features. However, frequent misclassifications between "REM" and "NREM" indicate shared physiological traits or feature extraction limitations.

**Analysis of a night of sleep** Fig 16 illustrates the predictions of a night's sleep for the same subject for sleep stages classification using original heart rate data and the ensemble

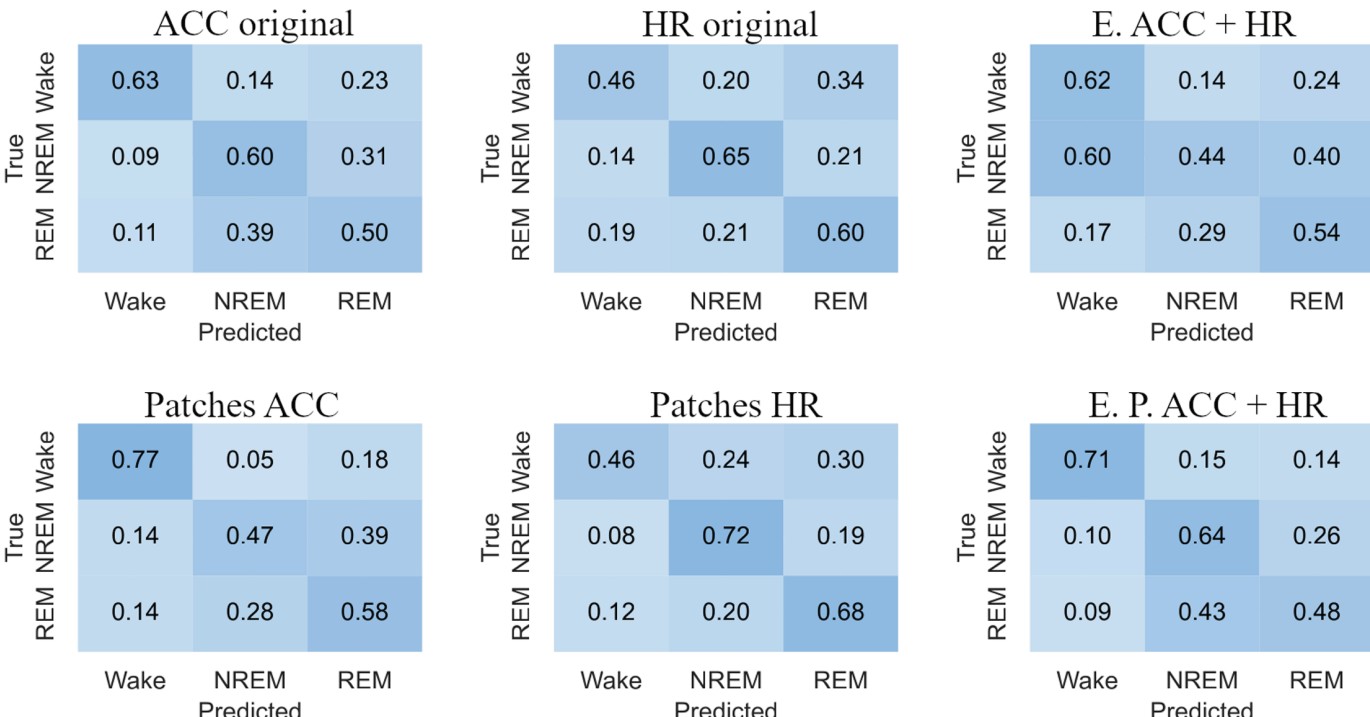

**Fig 13. GAF confusion matrices for sleep stages classification.** The highest balanced accuracy (62.18% ± 0.95%) was achieved with heart rate data and the ensemble of patches. The confusion matrix with heart rate patches demonstrated an increase in correct classifications of "NREM" and "REM". The most common misclassifications with accelerometer data were labeling "NREM" as "REM" and "REM" as "NREM". Meanwhile, with heart rate data, the most frequent error was classifying "Wake" as "REM".

of heart rate patches. By observing these figures, we can compare the best night's sleep for sleep stages classification (ensemble of heart rate patches using GAF) with the corresponding night's sleep using the original data.

With the original data, the most frequent errors are predictions of "Wake" for "NREM" and predictions of "REM" for "NREM." It is also noted that predictions of "Wake" are overestimated for the "REM" class. Analyzing the errors related to "NREM" predictions, the model confused this class more with "REM" than with "Wake." Finally, the least common misclassifications are "REM" and "Wake."

When analyzing the results of the ensemble of heart rate patches, a significant reduction in incorrect predictions of "Wake" for "NREM" is observed. The erroneous predictions of "NREM" to "REM" also showed a decrease. However, the classification errors of "REM" to "Wake" at the beginning of the night remained. Additionally, there is a notable occurrence of incorrect predictions of "NREM" to "Wake" around the 8-hour mark of the test.

## Other representations

To compare the visual representation methods of time series with the traditional approaches discussed in the related works, we used the raw data as input for 1D-CNN, LSTM, and GRU-based networks. Additionally, we compared the results of performing feature extraction from raw data as input for RF and Logistic Regression (LR).

**Raw data**  Models based on 1D-CNN, LSTM, and GRU were implemented directly on the raw data for comparison with visual representations. The network architecture is described by

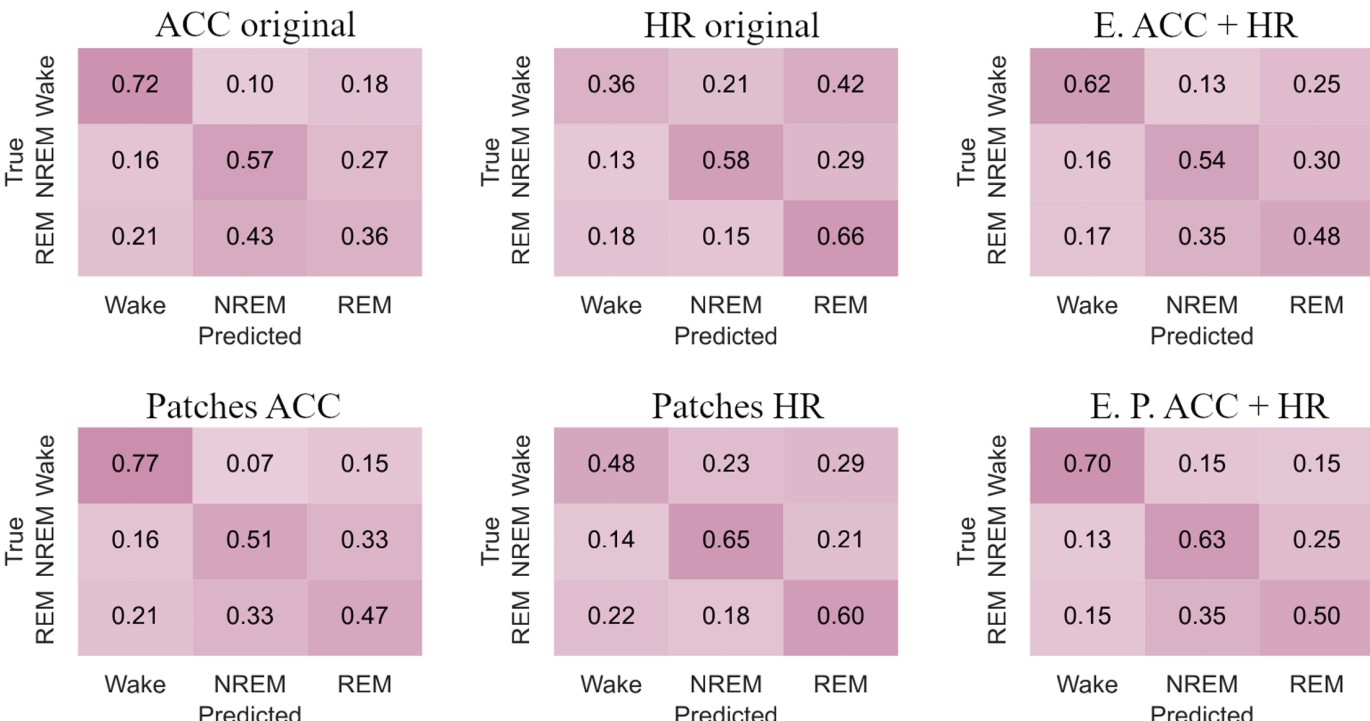

**Fig 14. MTF confusion matrices for sleep stages classification.** The highest balanced accuracy (60.97% ± 1.85%) was achieved with the ensemble combining accelerometer and heart rate patches. Accelerometer data and both types of ensembles most frequently classified "Wake" correctly, similar to other representations. For heart rate data, the incorrect classification of "Wake" as "REM" observed with original data decreased with the use of patches, resulting in a more balanced classification.

Mekruksavanich and Jitpattanakul [43] for HAR, with the first layer being a 1D convolution; the second, 1D *MaxPooling*; the third layer is another 1D convolution, followed by another *MaxPooling*; the penultimate layer is LSTM, and the last, a dense layer. To compare with models based on 1D-CNN and GRU, the penultimate layer of this architecture is replaced by a 1D-CNN and a GRU, respectively.

As with the visual representations, the networks received accelerometer data and heart rate data as input, and with the trained models, ensembles were performed using simple averaging, weighted averaging, and deep features to combine the results obtained with accelerometer data + heart rate data.

The results obtained for sleep/wake classification and sleep stages classification, with each configuration and each network, are presented in Tables 5 and 6.

For sleep/wake classification, similar to the visual representations, the accelerometer data present higher balanced accuracies than those obtained with heart rate data. The best results involve ensembles of both types of data for all networks. The highest balanced accuracy exceeds 75% with the GRU network and simple averaging ensemble. The best results obtained with only accelerometer or heart rate data were also achieved with GRU, exceeding 72% and 63%, respectively.

For sleep stages classification, it is possible to observe that, similar to the visual representations, the best-balanced accuracies were obtained with heart rate data, exceeding 57% using the LSTM network. The results obtained with ensembles did not show improvements for the

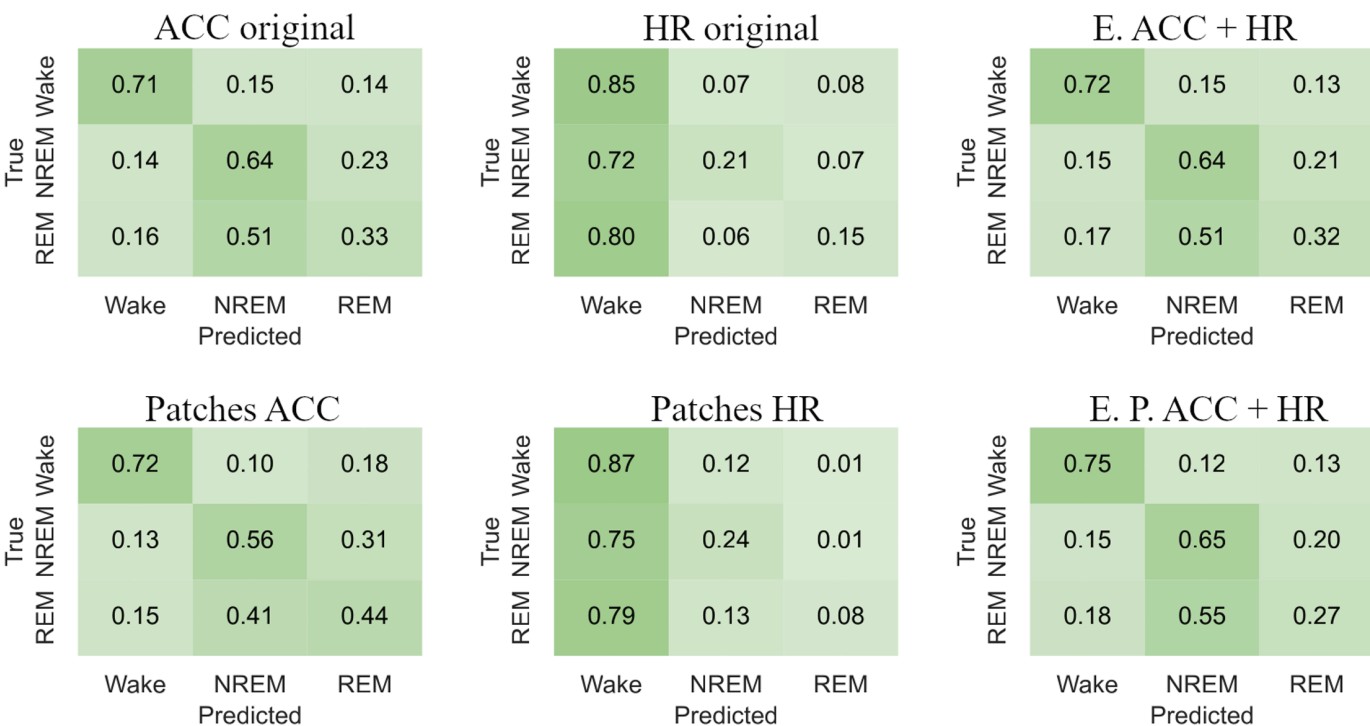

**Fig 15. Spectrograms confusion matrices for sleep stages classification.** The highest balanced accuracy (57.36% ± 2.68%) was achieved with heart rate data and the ensemble of patches. Heart rate data, both in its original and patched forms, resulted in fewer classifications of "NREM" and "REM," overestimating "Wake". However, the patched configuration increased the number of correct "REM" classifications while reducing incorrect "Wake" predictions. Both ensemble configurations improved correct classifications of "Wake" but showed increased confusion for "NREM" when the true class was "REM".

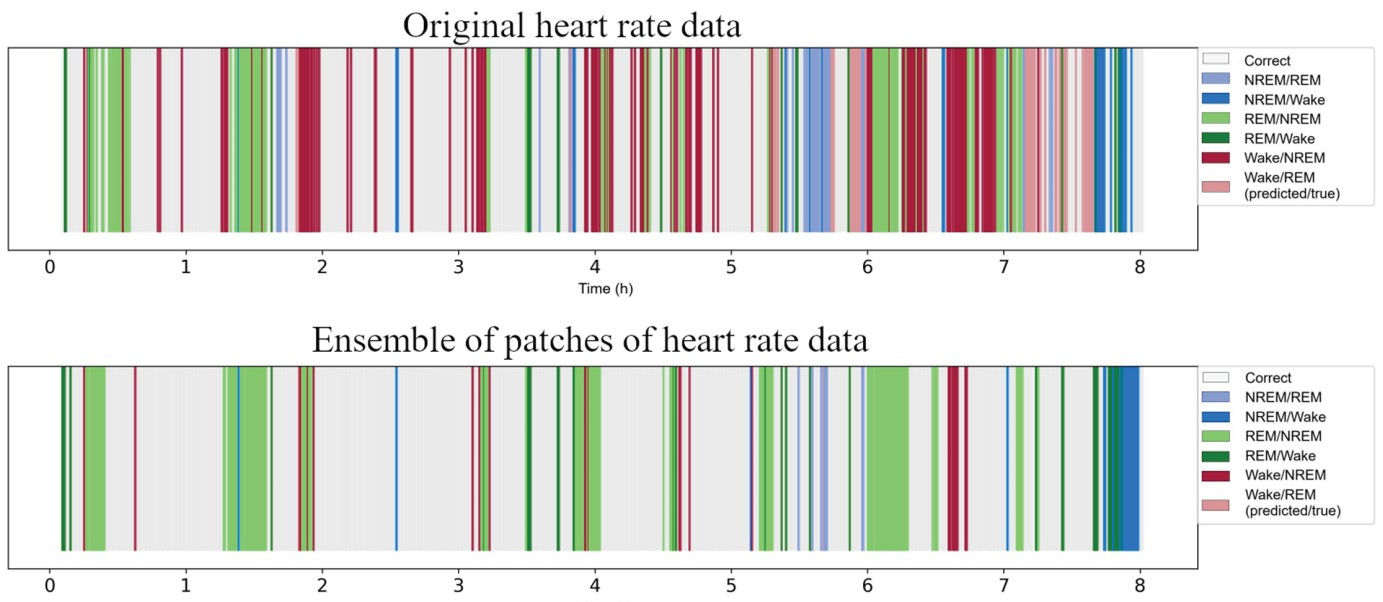

**Fig 16. Sleep stages classification over a night of sleep for a subject using the GAF representation.** Original data shows frequent "Wake" for "NREM" and "REM" for "NREM" errors. Ensemble of patches reduces these errors, but "REM" to "Wake" errors persist early on, and "NREM" to "Wake" errors appear around the 8-hour mark.

**Table 5.** Balanced accuracies obtained with raw data for sleep/wake classification.

| | | CNN | | LSTM | | GRU | |
|---|---|---|---|---|---|---|---|
| Config. | | ACC | HR | ACC | HR | ACC | HR |
| Without ensemble | | 71.60 | 52.00 | 71.55 | 56.29 | 72.54 | 63.88 |
| ACC + HR Ensembles | Simple Average | 71.30 | | 72.78 | | 75.62 | |
| | Weighted Average | 71.94 | | 72.84 | | 74.96 | |
| | Deep Features | 70.05 | | 71.91 | | 61.99 | |

Note: The underlined values represent the highest balanced accuracies for the corresponding data configurations.

**Table 6.** Balanced accuracies obtained with raw data for sleep stages classification.

| | | CNN | | LSTM | | GRU | |
|---|---|---|---|---|---|---|---|
| Config. | | ACC | HR | ACC | HR | ACC | HR |
| Without ensemble | | 46.66 | 46.70 | 47.87 | 57.55 | 46.84 | 56.03 |
| ACC + HR Ensembles | Simple Average | 45.66 | | 48.72 | | 45.08 | |
| | Weighted Average | 46.27 | | 48.42 | | 45.04 | |
| | Deep Features | 45.52 | | 51.75 | | 36.77 | |

Note: The underlined values represent the highest balanced accuracies for the corresponding data configurations.

1D-CNN and GRU networks. In contrast, the ensembles performed with the LSTM network improved the results presented with only accelerometer data.

**Features extraction** One could also wonder how good the classification would be if performing the characterization of the signal instead of using the raw data. Here, the feature extraction from accelerometer and heart rate data, as well as the performed ensemble, were based on the work of Walch et al. [25], which uses activity counts as a feature extracted from accelerometer data and local standard deviations extracted from heart rate data. The ensemble used the accelerometer data feature and the heart rate data feature as inputs to the models. Table 7 presents the balanced accuracies obtained for sleep/wake and sleep stages classifications.

Similar to the visual representation, the accelerometer data, compared to the heart rate data, present the best-balanced accuracies for sleep/wake classification, exceeding 76% with RF. For sleep stages classification, the best result exceeds 58% using RF and the feature ensemble.

**Comparison between representations** Table 8 compares balanced accuracies obtained with each data representation using an accelerometer, heart rate, and ensemble for sleep/wake classification. It can be observed that the visual representation achieves the best results in all cases, with an advantage of up to 5.8 percentage points with accelerometer data, 8.9 with heart rate data, and 4.8 with ensemble.

**Table 7.** Balanced accuracies obtained with feature extraction for sleep/wake and sleep stages classifications.

| | Sleep/wake classification | | | | Sleep stages classification | | | |
|---|---|---|---|---|---|---|---|---|
| | RF | | LR | | RF | | LR | |
| Config. | ACC | HR | ACC | HR | ACC | HR | ACC | HR |
| Without ensemble | 76.55 | 64.81 | 73.85 | 63.59 | 51.55 | 48.73 | 44.50 | 40.50 |
| ACC + HR Ensembles | 76.65 | | 73.98 | | 58.51 | | 46.51 | |

Note: The underlined values represent the highest balanced accuracies for the corresponding data/network configurations.

**Table 8**. **Comparison of the best-balanced accuracies obtained with different data representations for sleep/wake classification.**

|  | ACC | | | HR | | | ACC + HR Ensemble | | |
|---|---|---|---|---|---|---|---|---|---|
|  | Images | Raw Data | Features | Images | Raw Data | Features | Images | Raw Data | Features |
| Bal. ac. | 82.36 | 72.54 | 76.55 | 73.71 | 63.88 | 64.81 | 81.44 | 75.62 | 76.65 |
| Config. | GAF Patches | GRU | RF | GAF Patches | GRU | RF | GAF Patches | GRU | RF |

Note: The underlined values represent the highest balanced accuracies for the corresponding data type.

**Table 9**. **Comparison of the best-balanced accuracies obtained with different data representations for sleep stages classification.**

|  | ACC | | | HR | | | ACC + HR Ensemble | | |
|---|---|---|---|---|---|---|---|---|---|
|  | Images | Raw Data | Features | Images | Raw Data | Features | Images | Raw Data | Features |
| Bal. ac. | 60.66 | 47.87 | 51.55 | 62.18 | 57.55 | 48.73 | 61.48 | 51.75 | 58.51 |
| Config. | GAF Patches | LSTM | RF | GAF Patches | LSTM | RF | RP Patches | LSTM | RF |

Note: The underlined values represent the highest balanced accuracies for the corresponding data type.

Table 9 presents the best-balanced accuracies obtained with each type of representation using accelerometer, heart rate, and ensemble data and provides a comparison for sleep stages classification. As observed in the sleep/wake classification, the visual representation presents the best results in all cases. The difference in balanced accuracy using images reaches at least 9.1 percentage points with accelerometer data, 4.6 with heart rate data, and 3.0 with ensemble.

Although the use of images for data representation reduces the advantage when employing ensemble techniques compared to other forms of representation, it stands out by presenting significant gains in the isolated accelerometer data (leading to the best result for sleep/wake classification) and heart rate data (leading to the best result for sleep stages classification).

## Conclusions

Sleep stage classification is critical for evaluating sleep quality and identifying disorders. While PSG remains the gold standard, its high cost and requirement for controlled environments limit its accessibility. Smartwatches provide a practical alternative, but traditional methods, such as manual feature extraction for classical models and direct neural network application to raw data, face challenges related to noise, high dimensionality, and difficulty in capturing complex temporal patterns. This study investigated the use of visual representations of time series to enhance sleep stage classification using deep learning.

The results show that converting time series data into images allows the application of 2D-CNNs, which effectively capture spatial and temporal patterns. Among the tested visual representations, GAF achieved the highest performance, surpassing 82% balanced accuracy for sleep/wake classification and 62% for sleep stages classification when combined with patching and ensemble techniques. Compared to traditional approaches, visual representations outperformed raw data-based deep learning models and feature extraction techniques, with gains of up to 8.9 percentage points in sleep/wake classification and up to 9.1 percentage points in sleep stages classification.

Additionally, the study highlights the distinct contributions of accelerometer and heart rate data. Accelerometer data were more effective for sleep/wake classification, while heart rate data played a key role in distinguishing between sleep stages. The use of image patching and

ensembles improved classification performance by emphasizing local details (up to 3.7 percentage points for sleep/wake classification and up to 6.0 percentage points for sleep stages classification).

These findings have significant implications for sleep research and health monitoring. The proposed method enables more accurate sleep classification using affordable and widely available wearable devices. This could support large-scale sleep studies, early detection of sleep disorders, and personalized sleep improvement strategies. By providing a non-invasive alternative to PSG, this approach advances sleep research and may contribute to better health outcomes.

For future work, patches have shown promise in classifying sleep/wake states and sleep stages, suggesting exploring Transformer-based networks, such as the Vision Transformer (ViT) [44,45]. Normalization and filtering methods for accelerometer and heart rate data and post-processing techniques are crucial to improving data quality and classification accuracy. Temporal fusion for a night's sleep analysis is a future step for eliminating false positives, as it increases accuracy by integrating temporal patterns throughout the night, improving the detection of sleep stage transitions, and reducing false positives for more reliable sleep analysis. Additionally, explainability techniques are essential to make Deep Learning models more understandable and validatable by specialists [46].

Deploying Deep Learning technologies in wearable devices is challenging due to their computational and energy limitations [47]. Running these models and the processing of images demands substantial resources. Future research should prioritize optimizing visual representations to reduce computational costs without compromising performance, thereby enabling more efficient use of these technologies in wearables.

This study demonstrates that visual representations of time series data provide an effective alternative for sleep stage classification. These findings pave the way for advancements in wearable sleep monitoring and sleep disorder diagnosis.

## Supporting information

**S1 Table. Balanced accuracies obtained with each representation for sleep/wake classification.** Accelerometer data consistently outperformed heart rate data in all scenarios, with the GAF achieving the highest balanced accuracy (82.36% $\pm$ 3.24%) when using patch ensembles. Patch-based ensembles significantly improved balanced accuracy compared to original images.
(PDF)

**S2 Table. Balanced accuracies obtained with each representation for sleep stages classification.** Heart rate data often outperformed accelerometer data in balanced accuracies (except for the Spectrogram), with the GAF achieving the highest balanced accuracy (62.18% $\pm$ 0.95%) when using patch ensemble. Patch-based ensembles significantly improved balanced accuracy compared to original images.
(PDF)

## Author contributions

**Conceptualization:** Rebeca Padovani Ederli, Anderson Rocha, Zanoni Dias.

**Data curation:** Rebeca Padovani Ederli.

**Formal analysis:** Rebeca Padovani Ederli, Didier A. Vega-Oliveros, Aurea Soriano-Vargas, Anderson Rocha, Zanoni Dias.

**Investigation:** Rebeca Padovani Ederli, Didier A. Vega-Oliveros, Aurea Soriano-Vargas, Anderson Rocha, Zanoni Dias.

**Methodology:** Rebeca Padovani Ederli, Anderson Rocha, Zanoni Dias.

**Software:** Rebeca Padovani Ederli.

**Supervision:** Anderson Rocha, Zanoni Dias.

**Validation:** Rebeca Padovani Ederli, Didier A. Vega-Oliveros, Aurea Soriano-Vargas, Anderson Rocha, Zanoni Dias.

**Visualization:** Rebeca Padovani Ederli, Didier A. Vega-Oliveros, Aurea Soriano-Vargas.

**Writing – original draft:** Rebeca Padovani Ederli.

**Writing – review & editing:** Rebeca Padovani Ederli, Didier A. Vega-Oliveros, Aurea Soriano-Vargas, Anderson Rocha, Zanoni Dias.

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
