## [Decision Letter · Decision Letter 0]

22 Jan 2025

PONE-D-24-59060Time-series visual representations for sleep stages classificationPLOS ONE

Dear Dr. Padovani Ederli,

Thank you for submitting your manuscript to PLOS ONE. After careful consideration, we feel that it has merit but does not fully meet PLOS ONE’s publication criteria as it currently stands. Therefore, we invite you to submit a revised version of the manuscript that addresses the points raised during the review process.

We look forward to receiving your revised manuscript.

Kind regards,

Xiaohui Zhang

Academic Editor

PLOS ONE

Additional Editor Comments (if provided):

Reviewers' comments:

Reviewer's Responses to Questions

**Comments to the Author**

1. Is the manuscript technically sound, and do the data support the conclusions?

Reviewer #1: Yes

Reviewer #2: Yes

2. Has the statistical analysis been performed appropriately and rigorously? 

Reviewer #1: Yes

Reviewer #2: Yes

3. Have the authors made all data underlying the findings in their manuscript fully available?

Reviewer #1: Yes

Reviewer #2: Yes

4. Is the manuscript presented in an intelligible fashion and written in standard English?

Reviewer #1: Yes

Reviewer #2: Yes

5. Review Comments to the Author

Reviewer #1: The study explores the application of visual representations of time series data from smartwatches to classify sleep stages. The authors transform data collected from accelerometers and heart rate sensors into visual formats like Gramian Angular Fields (GAF), Recurrence Plots (RP), Markov Transition Fields (MTF), and spectrograms. These visual representations are processed using 2D convolutional neural networks (CNNs), with image patching and ensemble methods enhancing classification performance. The proposed method demonstrates superior accuracy compared to traditional raw data approaches. I have a few points that can help improve the manuscript.

1. EfficientNet is a class of models. The authors need to specify which specific EfficientNet model was used for the experiment.

2. The authors should clearly define the classification task they are working on. What exactly is the goal/benefit for different classification task, and what is the distribution of each class? Including a workflow or pipeline demonstration figure would greatly enhance clarity. For each task, the data distribution, including the actual number of cases(and their respective percentages), should be explicitly shown.

3. The authors mention a data imbalance issue. It would be helpful to know if they took any measures to address this problem or include it in a limitation discussion.

4. For cross-validation, it is unclear whether the authors used a stratified split based on the label of each sample. Clarifying this would be helpful. It is also unclear if the accuracy in the paper is the average accuracy of the 5-fold cross validation. If so, standard deviation should be included.

5. For better readability, I suggest placing the figure descriptions together with the figures.

6. The paper contains a large number of comparison results. The authors might consider highlighting the most significant ones and moving the rest to the supplementary material.

7. Conclusions and Discussion sections need more work. A summary of the study's goal should be included, along with how the findings support this goal. For example, how why is classifying sleeping stage important and how can the method developed benefit research or improve human well-being.

Reviewer #2: The paper utilizes Gramian Angular Fields, Recurrence Plots, Markov models, Transition Fields, and spectrograms to process time series data collected from smartwatches, aiming to enhance the classification of sleep stages. The analyses and experiments are rigorous. However, the methods utilized are largely standard, and the paper lacks a degree of innovation.

6. PLOS authors have the option to publish the peer review history of their article (what does this mean?). If published, this will include your full peer review and any attached files.

Reviewer #1: No

Reviewer #2: No

---

## [Author Response · Author response to Decision Letter 1]

26 Feb 2025

Paper title: Time-series visual representations for sleep stages classification

Journal: PLOS ONE

February 21th, 2025

Dear Editor and Reviewers,

We would like to thank the editor and reviewers for their comments and suggestions. They were valuable and certainly contributed to improving the quality of our manuscript. In this revised version, we addressed all points raised by the reviewers and made adjustments to enhance clarity, methodology description, and data presentation.

The comments, along with our continued research during the review process, allowed us to refine and strengthen our manuscript. The revised version has been submitted through the online submission system, with the main changes highlighted.

We appreciate the time and effort dedicated by the reviewers and the editor and hope that this version meets the journal’s criteria for publication. For any further questions, please do not hesitate to contact us.

Sincerely,

The authors

***

General Comment

After careful consideration, we feel that it has merit but does not fully meet PLOS ONE’s publication criteria as it currently stands. Therefore, we invite you to submit a revised version of the manuscript that addresses the points raised during the review process.

Response: We thank the editor and the reviewers for diligently dealing with our submission and for the very constructive feedback.

In this new version, we addressed the points of both reviewers, and we modified the manuscript according to their insightful and accurate comments. We are grateful to both reviewers and the editor for their positive comments and for the time dedicated to the reviewing process. With our revised manuscript, we send a point-by-point response in which we did our best to address all of them.

***

Reviewer #1

The study explores the application of visual representations of time series data from smartwatches to classify sleep stages. The authors transform data collected from accelerometers and heart rate sensors into visual formats like Gramian Angular Fields (GAF), Recurrence Plots (RP), Markov Transition Fields (MTF), and spectrograms. These visual representations are processed using 2D convolutional neural networks (CNNs), with image patching and ensemble methods enhancing classification performance. The proposed method demonstrates superior accuracy compared to traditional raw data approaches. I have a few points that can help improve the manuscript.

1. EfficientNet is a class of models. The authors need to specify which specific EfficientNet model was used for the experiment.

Response: We agree with the reviewer’s observation. To clarify, we now explicitly state that we used EfficientNet-B0 for our experiments. This information has been added to the “Materials and methods (Training and validation)” subsection.

***

2. The authors should clearly define the classification task they are working on. What exactly is the goal/benefit for different classification tasks, and what is the distribution of each class? Including a workflow or pipeline demonstration figure would greatly enhance clarity. For each task, the data distribution, including the actual number of cases (and their respective percentages), should be explicitly shown.

Response: Following the reviewer's suggestion, we have revised the manuscript to improve the clarity of our classification tasks. The “Introduction” section now explicitly describes the two classification tasks—two-stage (wake/sleep) and three-stage (wake/NREM/REM)—along with their practical relevance. The two-stage classification is useful for basic sleep detection, such as in large-scale sleep monitoring or initial sleep disorder screenings, while the three-stage classification enables detailed analysis of sleep architecture, essential for clinical applications like diagnosing sleep disorders or assessing sleep quality.

In addition, we have included Table 1 in the “Materials and Methods (Dataset)” subsection. This table provides the number of samples and percentages per class, highlighting the dataset imbalance.

To further enhance clarity, we also added Figure 2, which illustrates the overall workflow of the proposed methodology, including the transformation of time series into visual representations and subsequent classification steps.

***

3. The authors mention a data imbalance issue. It would be helpful to know if they took any measures to address this problem or include it in a limitation discussion.

Response: To mitigate the impact of class imbalance, we applied class weighting, where the loss function assigns higher weights to minority classes, ensuring a more balanced contribution during training. To address this comment, this strategy is now explicitly described in the “Materials and methods (Training and validation)” subsection.

***

4. For cross-validation, it is unclear whether the authors used a stratified split based on the label of each sample. Clarifying this would be helpful. It is also unclear if the accuracy in the paper is the average accuracy of the 5-fold cross validation. If so, standard deviation should be included.

Response: Following the suggestion, we have clarified these points in the revised manuscript. The “Materials and methods (Training and validation)” subsection now explicitly states that we used 5-fold cross-validation, ensuring that data from the same subject were never included in both training and validation within the same fold. Instead of a stratified split, we opted for a random split, allowing the model to capture the natural variability of sleep stage transitions, as now described in the “Materials and Methods (Training and Validation)” subsection.

Additionally, the “Materials and methods (Performance metrics and model evaluation)” subsection now specifies that the reported results correspond to the average balanced accuracy across the five folds, with the standard deviation included to quantify variability. Standard deviation values are reported in the “Results and discussion” section.

***

5. For better readability, I suggest placing the figure descriptions together with the figures.

Response: In the revised manuscript, figure descriptions have been placed together with their respective figures (figures 7-10 and figures 12-15).

***

6. The paper contains a large number of comparison results. The authors might consider highlighting the most significant ones and moving the rest to the supplementary material.

Response: We agree with the reviewer’s suggestion. To streamline the manuscript, we highlighted the most significant results in the main text and moved the full comparison tables (Tables 2 and 3) to the supplementary materials.

***

7. Conclusions and Discussion sections need more work. A summary of the study's goal should be included, along with how the findings support this goal. For example, why classifying sleeping stages is important and how can the method developed benefit research or improve human well-being.

Response: Following the reviewer’s suggestion, we revised the “Conclusion” section to better summarize the study's goal and clarify how the findings support it. The revised text now explicitly states that the objective was to enhance sleep stage classification using visual representations and deep learning. We also emphasize the importance of sleep stage classification for assessing sleep quality, detecting disorders, and advancing research.

***

Reviewer #2

The paper utilizes Gramian Angular Fields, Recurrence Plots, Markov models, Transition Fields, and spectrograms to process time series data collected from smartwatches, aiming to enhance the classification of sleep stages. The analyses and experiments are rigorous. However, the methods utilized are largely standard, and the paper lacks a degree of innovation.

Response: This study introduces a novel application for sleep stage classification using smartwatch data within a comprehensive full pipeline approach. While techniques such as Gramian Angular Fields (GAF), Recurrence Plots (RP), Markov Transition Fields (MTF) and spectrograms are established in other domains, their combined application and evaluation for this task represent a significant advancement. By assessing their effectiveness under identical conditions and comparing them to traditional approaches, this study bridges a critical gap that has not been addressed before in this context.

This study not only explores the application of these visual transformations but also in the integration of techniques such as image patching and ensemble methods, which significantly enhance classification performance. The combination of GAF with patching and ensemble techniques achieved a balanced accuracy of over 82% for two-stage classification (sleep/wake) and 62% for three-stage classification (wake/NREM/REM).

A direct comparison with other commonly used representations for sleep stage classification (feature extraction and direct use of raw data) highlights the advantages of visual representations, demonstrating improved classification accuracy and reinforcing their applicability in sleep monitoring, outperforming traditional methods by up to 8.9 percentage points in two-stage classification and 9.1 percentage points in three-stage classification. These results emphasize the ability of visual representations to capture fine-grained temporal structures and enhance prediction robustness.

Furthermore, this study provides novel insights into the distinct contributions of accelerometer and heart rate data. Accelerometer data proved more effective for distinguishing between sleep and wake states, while heart rate data played a critical role in three-stage classification. This finding underscores the complementary nature of these data sources and their importance for accurate sleep stage classification.

To address the reviewer's concern regarding innovation, we revised the “Introduction”, “Discussion”, and “Conclusion” sections to better emphasize the study's contributions. These revisions highlight how the proposed methodology advances the field by offering a non-invasive, cost-effective alternative to polysomnography and enabling more reliable health monitoring and early interventions. This study also paves the way for future research by demonstrating the potential of visual representations and deep learning techniques for sleep analysis.

---

## [Decision Letter · Decision Letter 1]

16 Mar 2025

PONE-D-24-59060R1Time-series visual representations for sleep stages classificationPLOS ONE

Dear Dr. Padovani Ederli,

Thank you for submitting your manuscript to PLOS ONE. After careful consideration, we feel that it has merit but does not fully meet PLOS ONE’s publication criteria as it currently stands. Therefore, we invite you to submit a revised version of the manuscript that addresses the points raised during the review process.

We look forward to receiving your revised manuscript.

Kind regards,

Xiaohui Zhang

Academic Editor

PLOS ONE

Journal Requirements:

Reviewers' comments:

Reviewer's Responses to Questions

**Comments to the Author**

1. If the authors have adequately addressed your comments raised in a previous round of review and you feel that this manuscript is now acceptable for publication, you may indicate that here to bypass the “Comments to the Author” section, enter your conflict of interest statement in the “Confidential to Editor” section, and submit your "Accept" recommendation.

Reviewer #1: All comments have been addressed

Reviewer #2: All comments have been addressed

2. Is the manuscript technically sound, and do the data support the conclusions?

Reviewer #1: Yes

Reviewer #2: Yes

3. Has the statistical analysis been performed appropriately and rigorously? 

Reviewer #1: Yes

Reviewer #2: Yes

4. Have the authors made all data underlying the findings in their manuscript fully available?

Reviewer #1: Yes

Reviewer #2: Yes

5. Is the manuscript presented in an intelligible fashion and written in standard English?

Reviewer #1: Yes

Reviewer #2: Yes

6. Review Comments to the Author

Reviewer #1: The study explores the application of visual representations of time series data from smartwatches to classify sleep stages. The authors transform data collected from accelerometers and heart rate sensors into visual formats like Gramian Angular Fields (GAF),Recurrence Plots (RP), Markov Transition Fields (MTF), and spectrograms. The visual representations are then feed into 2d neural network for classification. Most of the comments have been address.

On data split: Given the imbalanced distribution of the dataset (minority class only 7.7%), using random split instead of stratified split is not a good representation of the data distribution. Without stratification, a random split might lead to some folds having very few (or even no samples) from the minority class. Even with the weight adjustment on minority samples in training, this can still lead to unreliable performance estimates as train and test set may not have the same distribution. I suggest replacing the results with stratified split or checking the class distribution for the random split to make sure the data distribution in each fold is reliably representing the overall distribution.

Reviewer #2: All the review comments have been addressed, and the paper has been improved. Although I still believe the methods are standard and lack some innovation, the application and execution of these methods demonstrate creativity. There is potential for further development.

7. PLOS authors have the option to publish the peer review history of their article (what does this mean?). If published, this will include your full peer review and any attached files.

Reviewer #1: No

Reviewer #2: No

---

## [Author Response · Author response to Decision Letter 2]

21 Mar 2025

Paper title: Time-series visual representations for sleep stages classification

Journal: PLOS ONE

March 21th, 2025

Dear Editor and Reviewers,

We would like to thank the editor and reviewers for their constructive feedback and thoughtful suggestions. Their insights have been invaluable in improving the clarity and rigor of our manuscript.

In this revised version, we carefully addressed all the reviewers' comments. Specifically, we clarified methodological choices and provided additional justifications where necessary. We also analyzed the data split strategy in more detail to ensure the reliability of our results.

The revised manuscript has been submitted through the online submission system, with key changes highlighted. Below, we provide a point-by-point response to each reviewer’s comments, detailing the revisions made.

We greatly appreciate the time and effort the reviewers and the editor invested in evaluating our work. We hope that this version meets the high publication standards of PLOS ONE. After this round of revisions, we believe our manuscript is suitable for publication.

Please feel free to reach out for any further clarifications.

Sincerely,

The authors

General Comment

After careful consideration, we feel that it has merit but does not fully meet PLOS ONE’s publication criteria as it currently stands. Therefore, we invite you to submit a revised version of the manuscript that addresses the points raised during the review process.

Response: We thank the editor and the reviewers for diligently dealing with our submission and for the very constructive feedback.

In this new version, we addressed the points of both reviewers, and we modified the manuscript according to their comments. We are grateful to both the reviewers and the editor for their comments and the time they dedicated to the reviewing process. With our revised manuscript, we send a point-by-point response in which we did our best to address all of them.

***

Reviewer #1

The study explores the application of visual representations of time series data from smartwatches to classify sleep stages. The authors transform data collected from accelerometers and heart rate sensors into visual formats like Gramian Angular Fields (GAF),Recurrence Plots (RP), Markov Transition Fields (MTF), and spectrograms. The visual representations are then fed into a 2d neural network for classification. Most of the comments have been addressed.

On data split: Given the imbalanced distribution of the dataset (minority class only 7.7%), using random split instead of stratified split is not a good representation of the data distribution. Without stratification, a random split might lead to some folds having very few (or even no samples) from the minority class. Even with the weight adjustment on minority samples in training, this can still lead to unreliable performance estimates as the train and test set may not have the same distribution. I suggest replacing the results with stratified split or checking the class distribution for the random split to make sure the data distribution in each fold is reliably representing the overall distribution.

Response: We appreciate your insightful comment regarding the impact of using a random split instead of a stratified split in our cross-validation process. We acknowledge the concern that a random split may result in some folds having very few or no samples from the minority class, potentially affecting the reliability of performance estimates in particular scenarios.

To address this, we analyzed the class distribution across the five splits. We presented the results in the new Table 2 of the revised version (“Materials and methods / Training and validation” subsection). The table shows the percentage of Wake, NREM, and REM samples for each split in the training and validation sets. The training data maintains a stable class distribution across splits, ensuring a balanced representation during model learning.

While the validation distribution shows some variability—particularly in Split 1 and Split 2 for Wake and Split 5 for REM—this reflects real-world sleep data, where sleep stages are inherently imbalanced across different nights and individuals. Since the model is evaluated across multiple splits, the impact of these variations is minimized.

It is important to emphasize that our dataset consists of sleep time-series data, where each sample corresponds to data from a specific subject. To prevent data leakage, we ensured that data from the same subject were not simultaneously used for training and validation. Unlike traditional classification tasks where stratification can be applied at the sample level, our setup requires entire subjects to be assigned exclusively to either a training fold or a validation fold. Our decision to use a random split was also motivated by the goal of maintaining the natural variability of sleep stage transitions, which differ across nights and individuals.

Given the constraints imposed by subject-level separation, we believe that using a random split does not introduce substantial bias or compromise the reliability of our results in this study. Additionally, the data distribution in each fold reflects the overall distribution, enabling the model to be tested under conditions that closely resemble real sleep patterns.

While we appreciate the suggestion to replace the results with a stratified split, our analysis indicates that the observed stability in class distributions and the need to maintain subject independence make our current approach the most appropriate for this study.

We sincerely appreciate your valuable feedback. Please let us know if further clarification is needed.

***

Reviewer #2

All the review comments have been addressed, and the paper has been improved. Although I still believe the methods are standard and lack some innovation, the application and execution of these methods demonstrate creativity. There is potential for further development.

Response: We sincerely appreciate your time and effort in reviewing our paper. Your feedback has helped us refine our work, and we are grateful for your positive remarks regarding the application and execution of our methods. While we recognize that the techniques employed are well-established, we focused on demonstrating their effectiveness in a novel application. We agree that there is potential for further development, and we see this work as a foundation for future research.

Thank you again for your valuable insights and constructive evaluation.

---

## [Decision Letter · Decision Letter 2]

13 Apr 2025

Time-series visual representations for sleep stages classification

PONE-D-24-59060R2

Dear Dr. Padovani Ederli,

We’re pleased to inform you that your manuscript has been judged scientifically suitable for publication and will be formally accepted for publication once it meets all outstanding technical requirements.

Kind regards,

Xiaohui Zhang

Academic Editor

PLOS ONE

Additional Editor Comments (optional):

Reviewers' comments:

Reviewer's Responses to Questions

**Comments to the Author**

1. If the authors have adequately addressed your comments raised in a previous round of review and you feel that this manuscript is now acceptable for publication, you may indicate that here to bypass the “Comments to the Author” section, enter your conflict of interest statement in the “Confidential to Editor” section, and submit your "Accept" recommendation.

Reviewer #1: All comments have been addressed

2. Is the manuscript technically sound, and do the data support the conclusions?

Reviewer #1: Yes

3. Has the statistical analysis been performed appropriately and rigorously? 

Reviewer #1: Yes

4. Have the authors made all data underlying the findings in their manuscript fully available?

Reviewer #1: Yes

5. Is the manuscript presented in an intelligible fashion and written in standard English?

Reviewer #1: Yes

6. Review Comments to the Author

Reviewer #1: The study explores the application of visual representation of time series data from smartwatch to classify sleep stages. In the added table, the authors analyzed the class distribution across the splits and show that the training and testing maintain a relatively stable distribution. All the comments have been addressed and the paper has been improved.

7. PLOS authors have the option to publish the peer review history of their article (what does this mean?). If published, this will include your full peer review and any attached files.

Reviewer #1: No

---

## [Editor Report · Acceptance letter]

PONE-D-24-59060R2

PLOS ONE

Dear Dr. Padovani Ederli,

I'm pleased to inform you that your manuscript has been deemed suitable for publication in PLOS ONE. Congratulations! Your manuscript is now being handed over to our production team.

Kind regards,

on behalf of

Dr. Xiaohui Zhang

Academic Editor

PLOS ONE